

# Characterization the properties of VOCs and submicron organic aerosol at a street canyon environment

Sanna Saarikoski[1,*], Heidi Hellén[1], Arnaud P. Praplan[1], Simon Schallhart[1], Petri Clusius[2], Jarkko V. Niemi[3], Anu Kousa[3], Toni Tykkä[1], Rostislav Koutznetsov[1], Minna Aurela[1], Laura Salo[4], Topi Rönkkö[4], Luis M. F. Barreira[1], Liisa Pirjola[2,5], Hilkka Timonen[1]

[1]Atmospheric composition research, Finnish Meteorological Institute, Helsinki, 00101, Finland
[2]Institute for Atmospheric and Earth Systems Research, University of Helsinki, P.O. Box 64, 00014 Helsinki, Finland
[3]Helsinki Region Environmental Services Authority HSY, Helsinki, 00066, Finland
[4]Aerosol Physics Laboratory, Physics Unit, Tampere University, Tampere, 33014, Finland
[5]Department of Automotive and Mechanical Engineering, Metropolia University of Applied Sciences, P.O. Box 4071, 01600, Vantaa, Finland

*Correspondence to*: Sanna Saarikoski (sanna.saarikoski@fmi.fi)

**Abstract.** Urban air consists of a complex mixture of gaseous and particulate species from anthropogenic and biogenic sources that are further processed in the atmosphere. This study investigated the features and sources of volatile organic compounds (VOCs) and submicron organic aerosol (OA) at a street canyon in Helsinki, Finland in late summer. The main anthropogenic VOCs (aVOCs) and biogenic VOCs (bVOCs) were analyzed with an online gas chromatograph mass spectrometer, whereas the composition and size distribution of submicron particles was measured with a soot particle aerosol mass spectrometer.

This study showed that aVOC concentrations were significantly higher than bVOC concentrations at the street canyon. The largest aVOC concentrations were measured for toluene (campaign-average 1630 ng m$^{-3}$) and p/m xylene (campaign-average 1070 ng m$^{-3}$), while the dominating bVOCs was α-pinene (campaign-average 200 ng m$^{-3}$). The special focus of this research was also on the oxidation of VOCs and the association between VOCs and OA in ambient air. Production rates of the oxidized compounds (OxPR) from the VOC reactions revealed that the main local sources of the oxidation products were O$_3$ oxidation of bVOCs (66% of total OxPR) and OH radical oxidation of aVOCs and bVOCs (25 % of total OxPR). Overall, aVOCs produced much smaller portion of the oxidation products (18 %) than bVOCs (82 %). For particle phase organics, the source apportionment analysis extracted six factors for OA. Three OA factors were related to primary OA sources, traffic (24 % of OA, two OA types) and coffee roastery (7 % of OA), whereas the largest fraction of OA (69 %) consisted of oxygenated OA (OOA). OOA was divided into less oxidized semi-volatile OA (SV-OOA; 40 % of OA) and two types of low-volatility OA (LV-OOA; 30 %). SV-OOA was likely to originate from biogenic sources since it correlated with an oxidation product of monoterpene, nopinone. LV-OOA consisted of highly oxygenated long-range or regionally transported OA that had no correlation with local oxidant concentrations as it had already spent several days in the atmosphere before reaching the measurement site.

By investigating specific air quality cases, it was noticed that the influence of main pollutant sources was different for VOCs and OA at the street canyon. Vehicle emissions impacted both VOC and OA concentrations, whereas the influence of biogenic emissions was more clearly detected in the VOC concentrations than in OA due to the specific VOCs attributed to biogenic emissions. In contrast, the emissions from the local coffee roastery had a distinctive mass spectrum for OA but they could not be seen in the VOC measurements due to the measurement limitations for the large VOC compounds. Long-range transport increased the OA concentrations and oxidation



state considerably, while its effect was observed less clearly in the VOC measurements due to the oxidation of most VOC in the atmosphere during the transport. Overall, this study revealed that in order to properly characterize the impact of different emission sources to air quality, health and climate, it is of importance to describe both gaseous and particulate emissions and understand how they interact as well as their phase transfers in the
atmosphere during the aging process.

# 1 Introduction

Anthropogenic air pollution is one of the greatest environmental issues with broad impacts on air quality, climate and health (Lelieveld et al., 2015; Schraufnagel, 2020; IPCC, 2021). Impaired air quality has been estimated to be responsible for a large portion of annual morbidity and mortality around the world due to several respiratory,
cardiovascular, immune, and nervous system diseases (e.g. Dominici et al. 2006; Genc et al., 2012; Glencross et al., 2020). In 2019, air pollution accounted for an estimated 6.7 million deaths, about 12 % of all deaths registered in the same year (Brauer et al., 2021). The spatial and temporal variation of gaseous and particulate pollutants from anthropogenic sources depends largely on source type, atmospheric lifetime of pollutants and local meteorology. Biogenic emissions also play an important role in atmospheric pollution. For example, it has been
shown that biogenic precursors can form secondary organic aerosol (SOA) which can be enhanced by the presence of anthropogenic pollutants such as nitrogen oxides ($NO_x$) and sulfur dioxide ($SO_2$) (Edney et al., 2005; Kroll et al., 2006; Budisulistiorini et al., 2015).

Volatile organic compounds (VOCs) are particularly important atmospheric gaseous pollutants. Globally, anthropogenic VOCs (aVOCs) contribute approximately 14 % to the total VOC emission, while the contribution
of biogenic VOCs (bVOCs) is over 80 % (Guenther et al., 2012; Crippa et al., 2020). In urban environments, the contribution of aVOCs is much larger. While traffic has been historically the main contributor to aVOCs, its significance is decreasing in developed countries due to the imposed regulations (e.g. EEA, 2019). Therefore, other VOC sources, like the use of volatile chemical products (VCPs), are becoming more important and are suspected to be responsible for already half of the aVOC emissions in urban areas (McDonald et al., 2018). This
can be seen in the study of Karl et al. (2018), where VOC fluxes were measured above a city. From their measurements, several sources of VCPs were identified, e.g. cleaning agents, paint, human emissions (skin), healthcare products and disinfectants. The main sources of bVOCs in urban environments are green urban infrastructure (e.g. parks, green roofs, forests), which are used in many cities not only as recreation zones but also for heat and air pollution mitigation and water interception (Livesley et al., 2016; Fitzky et al., 2019). However,
it should be noted that also some VCPs can be included in bVOCs as e.g. cleaning agents and personal care products contain the same compounds as naturally emitted bVOCs. Once emitted into the atmosphere, VOCs get oxidized by either the hydroxyl radical (OH), ozone ($O_3$), or the nitrate radical ($NO_3$). The oxidation products vary greatly depending on the VOC composition and atmospheric conditions, but often they have a lower volatility than their precursors and are potential contributors to the formation and/or growth of SOA.
SOA can be produced from both aVOCs and bVOCs via new particle formation or the condensation of oxidation products on existing particles. The volatility and oxidation of organics continues further in the particle phase with photochemical processing. For example, in a study performed in Mexico City, semi-volatile oxygenated organic aerosol (SV-OOA) was the dominant OA type but the oxygen to carbon ratio (O:C) and the contribution of low-



volatility oxygenated organic aerosol (LV-OOA) increased with OA aging (Jimenez et al., 2009). Similar
transformation has also been observed in the laboratory studies. SOA formed from the oxidation of α-pinene
became more similar to ambient SV-OOA after some aging, and then with continued oxidation, evolved to be
similar to ambient LV-OOA (Jimenez et al., 2009). These results suggest that oxygenated organic aerosol (OOA)
components become more chemically similar with photochemical aging regardless of the original source of OOA.
In addition to secondary production, OA can also be emitted directly from the sources (primary OA, POA) The
main sources of POA in urban areas are traffic, residential biomass combustion, industry and energy production
(e.g. Crippa et al., 2014; Timonen et al. 2013; Zhang et al., 2019). However, it should be noted that these sources
emit also gaseous precursors for SOA.

To understand the secondary aerosol formation processes in urban areas, detailed information about the chemistry
of both gaseous compounds, primary and secondary particulate species as well as local meteorology are needed.
Harrison (2018) underlined that urban environments usually have high levels of primary emissions with strong
concentration gradients as mixing processes are heavily influenced by the presence of buildings and potentially
by the urban heat island. Reaction timescales are therefore shorter in urban areas compared to the well-mixed
regional atmosphere. Kim et al. (2018) found that in Seoul, Korea the formation of LV-OOA and sulfate was
mainly promoted by elevated ozone concentrations and photochemical reactions during daytime, whereas SV-
OOA and nitrate formation were attribute to both nocturnal processing and daytime photochemical reactions. Yu
et al. (2019) identified three bVOCs (α-pinene, limonene, and camphene) and one aVOC (styrene) as the possible
key VOC precursors to particulate organic nitrates in the megacity of Shenzhen, China. Sjostedt et al. (2011)
concluded that biogenic precursors contribute significantly to the total amount of SOA formation, even during
periods of urban outflow. They found that the importance of aromatic precursors was more difficult to assess
given that their sources are likely to be localized and thus of variable impact at the sampling location.

The aim of this study was to investigate the characteristics and sources of VOCs and particulate submicron (< 1
µm in diameter) OA at a street canyon in late summer. For the first time in Helsinki, a wide range of aVOCs and
bVOCs was analyzed with an online gas chromatograph mass spectrometer (GC-MS). The 1-hour time-resolution
for the VOC data enabled the study of short-term variability of the concentrations and allowed for comparisons
with particle measurements conducted with a real-time aerosol mass spectrometer (AMS). The OA mass spectra
from the AMS was further analyzed by positive matrix factorization (PMF) for the sources and properties of OA.
The specificity of the AMS source apportionment and VOC concentrations to various urban sources were
examined by selecting four time periods that were examined in detail. Moreover, the oxidation of VOCs was
investigated thoroughly by calculating the production rates of the VOC oxidation products, and their contribution
to SOA formation was assessed. This study provides novel information on the sources of anthropogenic and
biogenic VOCs and OA in an urban environment and elucidate atmospheric oxidation processes and SOA
formation in a street canyon. This information is currently highly needed by air quality authorities and modelers
all over the world to improve urban air quality as well as the models for aerosol dynamics and atmospheric
chemistry.

## 2 Experimental methods

### 2.1 Measurement site



The measurement campaign was conducted from 14 August to 13 September 2019 at the Helsinki Supersite measurement station (street address Mäkelänkatu 50), Finland. The station is located at the kerbside of the street and is maintained by the Helsinki Region Environmental Services Authority (HSY). The street consists of six

lanes for motorized traffic, two rows of trees, two tram lanes and two sidewalks, for a total width of 42 m (Hietikko et al., 2018). Mäkelänkatu is one of the busiest traffic sites in the Helsinki city center with a traffic density of about 28 000 vehicles per weekday with a heavy-duty vehicle share of 10 % (statistics from the City of Helsinki). Long-term concentrations, composition and trends of submicron particulate matter (PM$_1$) at the Helsinki Supersite have been presented in Barreira et al. (2021).

### 2.2 Instruments

#### 2.2.1 Online TD-GC-MS

The concentrations of VOCs were measured with an in situ thermal desorpter - gas chromatograph- mass spectrometer (TD-GC-MS, Perkin Elmer Inc., Waltham, US). Studied compounds were hydrocarbons with 5 to 15 carbon atoms, which are known to be important SOA precursors. Based on their most probable origin,

compounds were classified as aVOCs and bVOCs even though some bVOCs are also known to have anthropogenic sources. The studied aVOCs consisted of aromatic hydrocarbons (benzene, toluene, ethylbenzene, p/m-xylene, styrene, o-xylene, 3-ethyltoluene, 4-ethyltoluene, 1,3,5-trimethylbenzene, 2-ethyltoluene, 1,2,4-trimethylbenzene and 1,2,3-trimethylbenzene). Analyzed bVOCs were isoprene, monoterpenoids (α-pinene, camphene, β-pinene, Δ3-carene, p-cymene, 1,8-cineol and limonene), sesquiterpenes (longicyclene, iso-

longifolene, β-caryophyllene and α-humulene) and an oxidation product of β-pinene (nopinone).

VOCs were collected into the cold trap (Tenax TA 60-80/ Carbopack B 60-80) of the thermal desorption unit (TurboMatrix 350, Perkin-Elmer Inc., Waltham, US) connected to a gas chromatograph (Clarus 680, Perkin-Elmer Inc., Waltham, US) coupled to a mass spectrometer (Clarus SQ 8 T, Perkin-Elmer Inc., Waltham, US). The hydrophobic cold trap was kept at 25 °C for the removal of humidity. 30-minute samples were taken with a 1-

hour time-resolution and a flowrate of 40 mL min$^{-1}$. The main flow going to the instruments through FEP tubing (ca. 5 m length, i.d. 1/8″) was approximately 0.8 L min$^{-1}$. To also enable the measurements of highly ozone reactive terpenes, a heated stainless steel tube was connected to the main flow path to remove ozone before sampling (see Hellén et al., 2012a). For calibration, standards were injected as methanol solutions into sorbent tubes (Tenax TA 60-80/Carbopack B 60-80), methanol was flushed away in nitrogen (6.0) flow and the tubes

were thermally desorbed and analyzed as samples. Five-point calibration curves were used. For isoprene calibration, a gas standard from National Physical Laboratories (UK) was used. The method has been described in detail by Helin et al. (2020).

#### 2.2.2 SP-AMS

Size-resolved chemical composition of submicron particles i.e. organics, sulfate, nitrate, ammonium, chloride and

refractory black carbon (rBC) was determined with a soot particle aerosol mass spectrometer (SP-AMS, Aerodyne Research Inc., Billerica, US, Onasch et al., 2012). The SP-AMS collected data with two-minute time-resolution of which half of the time the instrument operated in a mass spectra mode (mass concentrations) and half of the time in a particle time-of-flight (PToF) mode (mass size distributions). The measured particle size range of the SP-AMS is roughly from 40 nm to 1 µm. A collection efficiency (CE) of one was applied to the data as with this



value the total mass from the SP-AMS was comparable with that from the differential mobility particle sizer (DMPS, Sect. 2.2.3) operating at the site. A relative ionization efficiency (RIE) of 0.1 was used for rBC based on the calibration with Regal black (REGAL 400R pigment black, Cabot Corp.). However, due to the considerable uncertainties related to the quantification of rBC with the SP-AMS (e.g. imperfect laser beam alignment), black carbon (BC) concentrations presented in this paper are taken from the aethalometer (Sect. 2.2.3). The SP-AMS

data was analysed with IGOR 6.37 SQRL 1.62A and PIKA 1.22A software.

### 2.2.3 Aethalometer, DMPS and auxiliary measurements

Equivalent black carbon (eBC) measurements were conducted using a dual-spot aethalometer (AE33, Aerosol d.o.o., Ljubljana, Slovenia), which allows real-time measurement of aerosol light absorption at 7 wavelengths (370–950 nm; Drinovec et al., 2015). The sampling flow rate was set to 5 L min$^{-1}$ and the inlet cut-off size was 1

μm (sharp cut cyclone, BGI model SCC1.197). The time-resolution was one minute. The filter tape was a M8060 and consisted of TFE-coated glass fiber filters. BC concentrations from wood burning ($BC_{wb}$) and fossil fuel ($BC_{ff}$) were estimated using the aethalometer model (Sandradewi et al., 2008). Absorption Ångström exponents of 1.1 and 1.6 were applied to fossil fuel ($\alpha_{ff}$) and wood burning ($\alpha_{wb}$), respectively, as those values have been previously optimized for the street canyon site (Helin et al., 2018).

Submicron particle number size distributions were measured using a DMPS (Knutson and Whitby, 1975). The DMPS includes a differential mobility analyzer (DMA, Vienna-type), used for particle sizing, and a condensation particle counter (CPC, A20 Airmodus, Helsinki, Finland) for obtaining particle number concentrations for each size bin. The time-resolution of the DMPS was 9 minutes and the scanned particle size range was 6 to 800 nm (mobility diameter, $D_m$), but due to a power source issue, the three smallest stages of the DMPS were excluded

from data and the size distribution was calculated only for the size range of 10–800 nm. The DMPS size distribution was compared to that of the electrical low pressure impactor (ELPI+, Dekati Ltd., Tampere, Finland, Järvinen et al., 2014) which operated at the site during the last week of the measurement campaign (5–12 September 2019). The number size distributions from the DMPS and ELPI were similar indicating that the DMPS data was reliable after excluding the smallest stages of the DMPS. The DMPS number size distribution was

converted to the mass size distribution by assuming spherical particles and a particle density of 1.42 g cm$^{-3}$ which has been shown to be the average density of submicron particles at the site (Barreira et al., 2021).

Basic air quality parameters were also measured at the site. The concentration of NO$_x$ was measured by an APNA-370 analyzer (Horiba, Kyoto, Japan), O$_3$ by using an ambient O$_3$ monitor (APOA-370, Horiba, Kyoto, Japan), CO by APMA-360 (Horiba, Kyoto, Japan) and PM$_{2.5}$ and PM$_{10}$ concentrations by a tapered element oscillating

microbalance (1405 TEOM™, Thermo Fischer Scientific, Waltham, US) with a time-resolution of one minute. The mass concentration of coarse particles (PM$_{2.5–10}$) was calculated by subtracting PM$_{2.5}$ from PM$_{10}$. Of meteorological parameters, temperature (T), relative humidity (RH) and precipitation were measured at the street canyon site, while wind speed and wind direction were measured at a meteorological station above the roof level (53 meters above the land surface) located approximately 900 m north-west from the measurement site. The

mixing height was calculated using the model (MPP-FMI) presented by Karppinen et al. (2000). In order to investigate a long-range transport (LRT) episode detected at the site on 9–11 September 2019, the origins of the air masses were calculated using the NOOA HYSPLIT model (Stein et al. 2015; Rolph et al. 2017). 96-hour back trajectories were calculated for every hour at the height of 200 m above sea level.





### 2.3 Data analysis

#### 2.3.1 Calculation of the production rates of oxidized compounds

Production rates of oxidized compounds (OxPRs) from VOCs $i$ were calculated from their concentration, the concentration of the oxidant, and their respective reaction rate:

$$OxPR = \frac{d[products]}{dt} = \sum[VOCi]\left(k_{OH+VOCi}[OH] + k_{O_3+VOCi}[O_3] + k_{NO_3+VOCi}[NO_3]\right) \qquad (1)$$

where $k_i$ is the reaction rate coefficient of a VOC with an oxidant (OH, $O_3$ or $NO_3$) and [$VOC_i$] is the concentration of corresponding VOC or oxidant. Details of the reaction rate coefficients used in this study can be found in Table S1. Concentrations of $O_3$ were from the local measurements, while OH and $NO_3$ radical concentrations were modelled using the ARCA box model as described in Sect. 2.4.

#### 2.3.2 PMF for the SP-AMS data

The SP-AMS dataset was analyzed for the sources and types of OA with a positive matrix factorization algorithm (CU AMS PMF tool v. 2.08D, Paatero and Tapper, 1994; Ulbrich et al., 2009). The number of factors was varied from 2 to 8, and the solution obtained with 6 factors provided the most reasonable results. The factors were identified as two hydrocarbon-like OA (HOA) factors referred to as HOA-1 and HOA-2, one semi-volatile oxygenated OA factor (SV-OOA), one low-volatility oxygenated OA factor (LV-OOA), one LV-OOA factor from long-range transport (LV-OOA-LRT), and a coffee roastery OA factor (CoOA). HOA-1 and HOA-2 correlated only moderately in terms of time series (r = 0.42) and mass spectra (r = 0.69), and therefore, they were supposed to represent different types of OA and were not combined. The results for 7 and 8 factors did not provide any additional information; in the 7-factor solution, HOA-2 was split further in two factors, whereas in the 8-factor solution also LV-OOA-LRT was divided into two identical factors. Two periods of very high OA concentrations were excluded from the PMF data matrix. Those periods were 1) from 2:00 to 4:15 on 31 August 2019 (Saturday), and 2) from 23:00 on 31 August 2019 to 1:20 on 1 September 2019 (Saturday-Sunday night). During those periods OA consisted purely of hydrocarbon fragments (similar to the HOA-1 factor), but when those cases were included in the data set, they distorted the calculation of the campaign- and diurnal averages as well as the PMF analysis. The source for the high HOA-1 concentrations was not found, but since CO, $CO_2$ or $NO_x$ concentrations did not increase during those periods, the source was not likely to be any typical combustion process.

Besides OA, PMF was also applied to the mass spectra of organics accompanied by $NO^+$ and $NO_2^+$ ions to explore the presence of organonitrates in the mass spectra of the PMF factors. As in the OA PMF analysis, PMF was run with up to 8 factors with $NO^+$ and $NO_2^+$ ions (hereafter called OA + $NO^+/NO_2^+$ solution). The 7-factor OA + $NO^+/NO_2^+$ solution corresponded closely to the 6-factor solution with OA since the seventh factor in the OA + $NO^+/NO_2^+$ solution represented inorganic ammonium nitrate, consisting mostly of $NO^+$ and $NO_2^+$ ions and contributing only 1 % to the total OA signal. The comparison of the PMF solutions for OA (6-factor solution) and OA + $NO^+/NO_2^+$ (7-factor solution) in terms of high-resolution mass spectra and mass concentrations is presented in Figs. S1 and S2. For the POA factors (HOA-1, HOA-2, CoOA), the correlation was very good for both mass spectra and mass concentrations, while for the oxygenated OA factors (especially for LV-OOA and LV-OOA-LRT) there were small differences between the solutions. The OA + $NO^+/NO_2^+$ solution was utilized only to assess





the contribution of organonitrates to the PMF factors (Sect. 3.4.2), and all the other data shown in this paper was obtained from the PMF solution for OA (OA solution with 6 factors).

**2.4 Air chemistry modelling with ARCA box**

The Atmospherically Relevant Chemistry and Aerosol box model (ARCA box; Clusius, 2020) was used to estimate the concentrations of OH and $NO_3$. ARCA box combines the most recent development in terms of atmospheric modelling, including the latest master chemical mechanism (MCM) version (http://mcm.york.ac.uk/), complemented by the peroxy radical autoxidation mechanism (PRAM; Roldin et al., 2019), as well as atmospheric cluster dynamics code (ACDC; McGrath et al., 2012) for molecular clustering and representation of aerosol particle size distribution and its evolution.

Six periods from the campaign period during which VOC measurements were available were simulated in ARCA box (v1.2.0) with a 1-hour time-resolution. The input for the model consisted of in situ measurements of meteorological parameters (temperature, pressure, relative humidity), trace gas concentrations (NO, $NO_2$, $O_3$, CO), and VOC concentrations (benzene, toluene, xylenes, ethylbenzene, ethyltoluenes, trimethylbenzenes, styrene, isoprene, pinenes, limonene, carene, and β-caryophyllene). In addition, global irradiance from the SMEAR III station located ca. 940 m to the north-east was used (https://smear.avaa.csc.fi/), as well as $SO_2$ concentrations from the urban site Kallio, about 1.0 km south of the street canyon site. The surface albedo, used in calculating the actinic flux from the measured irradiance, was set to 0.2. The modelled concentrations were linearly interpolated to match the times of the VOC measurements for the calculation of OxPRs.

In the present study, new particle formation and coagulation was not simulated. The particle size distribution measured at the site was used to calculate the condensation sink and oxidation products which were allowed to condense on the particles.

**3 Results and discussion**

**3.1 General description of the measurement period**

**3.1.1 Meteorology and inorganic gases**

The measurement period from 14 August to 13 September 2019 was characterized by a warm late summer and early autumn weather. The temperature was on average 17 °C with a clear variation between daytime (maximum of 25 °C, minimum 16.5 °C) and night-time (maximum of 17.5 °C, minimum 9.3 °C) (Fig. S3). There was rain on a total of 15 days with the maximum rainfall observed on 23 August. Wind speed varied from 0 to 10.5 m s$^{-1}$ with an average of 4.4 m s$^{-1}$. During the campaign, the dominant wind direction was from the south to south-west sector, and consequently, the measured concentrations were likely to be impacted by the emissions from Central Europe. Moreover, there was a distinctive LRT pollution episode between 9 and 11 September, and based on the air mass trajectories, the air masses originated from Central Europe and Russia during that period. This LRT period has been studied earlier in Salo et al. (2021) in terms of the lung deposited surface area (LDSA) of particles. In this paper, the impact of LRT period on VOC and particle chemistry will be discussed in detail in Sect. 3.3.3.

For the inorganic gases, the campaign-average NO, $NO_2$, $NO_x$, $O_3$ and CO concentrations were 18.9, 30.3, 59.4, 45.1 and 200 µg m$^{-3}$, respectively. As expected for a traffic environment, NO, $NO_2$ and $NO_x$ had a clear daily variation displaying a maximum during morning traffic (7:00–9:00) and a second, but less pronounced, peak in





the afternoon (15:00–17:00). $O_3$ had an opposite diurnal trend to $NO/NO_2/NO_x$ with a minimum in the morning (7:00–9:00). CO was slightly elevated in daytime with an increase of ~50 µg m$^{-3}$ compared to the night-time concentrations. The concentrations and diurnal patters of inorganic gases will be discussed more in the following sections.

### 3.1.2 Particle number, mass and chemical composition

The average particle number concentration for > 10 nm particles was 9200 particles cm$^{-3}$. Particle number concentration followed the traffic pattern having the largest concentrations during the morning rush hour (~17 000 particles cm$^{-3}$) and the smallest concentrations (~4800 particles cm$^{-3}$) during the early morning hours (~3:00–5:00) (Fig. S4). On average, 51 % of the particles were in the size range of 10–25 nm with the 10–25 nm fraction being smallest in early morning (38 %). Particle number and mass size distributions will be discussed later in terms of specific sources (Sect. 3.3).

The average mass concentrations of fine (PM$_{2.5}$) and coarse (PM$_{2.5–10}$) particles were virtually equal (6.8 and 6.7 µg m$^{-3}$, respectively), but there was more variation in the PM$_{2.5–10}$ than in the PM$_{2.5}$ concentration during the campaign (Fig. S4). Both PM$_{2.5}$ and PM$_{2.5–10}$ had elevated concentrations during the day and the smallest concentrations in early morning hours, similar to the number concentrations. Based on the DMPS data, the average mass concentration of PM$_1$ was 3.7 µg m$^{-3}$ (calculated with the density of 1.42 g cm$^{-3}$) while the sum of the SP-AMS species and eBC from the aethalometer was slightly larger being on average 4.5 µg m$^{-3}$.

PM$_1$ particles consisted mostly of OA (53 %) followed by eBC (30 %). The average contributions of inorganic species were small being 11, 2.8, 3.2 and 0.25 % for sulfate, nitrate, ammonium and chloride, respectively. Compared to the average composition at the site presented in Barreira et al. (2021), the contribution of eBC was larger and the contributions of inorganic species were smaller in this study. Larger eBC can be explained at some extent by the use of a multi-angle absorption photometer (MAAP) in Barreira et al. (2021), as the MAAP gave approximately 72 % of the AE33 values at a street canyon site (Helin et al., 2018). The eBC concentrations followed the traffic pattern with a maximum in the morning and a smaller concentration peak in the afternoon during the evening traffic. OA had slightly larger concentrations before noon, but besides that, there was no clear diurnal trend for OA. In terms of inorganic species, nitrate had smaller concentrations in daytime indicating its semi-volatile characteristic. Sulfate, ammonium and chloride did not display any diurnal pattern.

### 3.1.3 Volatile organic compounds

Anthropogenic VOCs had clearly higher concentrations (campaign-average 4.8 µg m$^{-3}$) than biogenic VOCs (campaign-average 0.57 µg m$^{-3}$) with toluene and p/m-xylene being most abundant aVOCs (Table S1). Previous source apportionment studies conducted in Helsinki in early 2000s indicated that traffic was clearly the most important source of aVOCs (Hellén et al. 2006). However, since then, the traffic emissions have decreased due to the emission regulations (e.g. EEA, 2019), and therefore, the relative importance of other sources (e.g. VCPs) might have increased (McDonald et al. 2018). aVOC concentrations were highest during the rush hours, with the morning peak being more intense than the evening peak (Fig. 1a). This is possibly due to a lower mixing layer height and therefore less dilution in the morning (Fig. 1b). Also the direction of vehicles depends on the time of day since in the morning there is more traffic in the lanes close to the site (southbound, towards the city center), whereas in the evening there is more traffic towards the north using the lanes further from the site. Styrene had a





different diurnal variation from all the other VOCs as it is only aVOC having significant reactions with ozone. The measured average benzene concentration ($0.34 \pm 0.220$ µg m$^{-3}$) was well below the lowest annual average concentration threshold (2 µg m$^{-3}$) given by EU (EU, 2008). Usually, the highest concentrations of aromatic hydrocarbons are measured in the winter due to longer lifetimes and higher emissions in the winter (Hellén et al.

2012b).

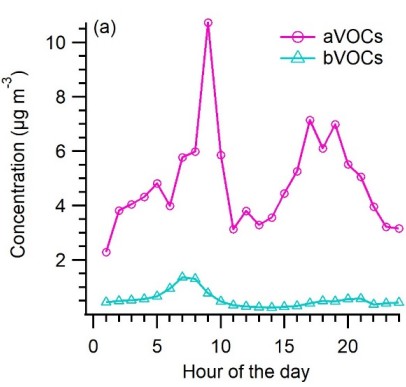
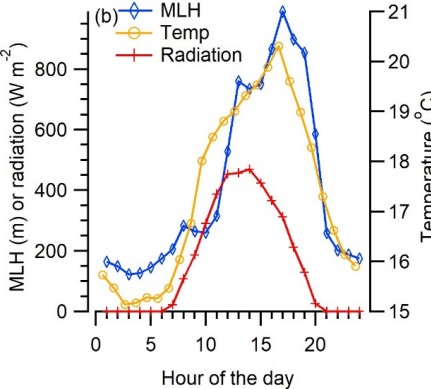

**Figure 1**. Campaign-average diurnal variation of aVOC and bVOC concentrations (a) and the average mixing layer height (MLH), ambient temperature (Temp) and solar radiation (Radiation) during the VOC measurements (b).


Most bVOC concentrations (Table S1) were well above the detection limits during these measurements in late August/early September even though generally bVOC emissions and concentrations are known to be highest during the main growing season in July/early August (Tarvainen et al., 2007; Hellén et al., 2018). Since the emissions from vegetation are known to be temperature and light dependent, the relatively high ambient

temperature during the measurements at least partly explains the high bVOC concentrations (Fig. 1). Of bVOCs, monoterpenes had the highest concentrations, α-pinene being the most abundant. Isoprene concentrations were clearly lower than those of monoterpenes. This was expected since the most common trees in Finland are known to be mainly mono- and sesquiterpene emitters (e.g. scots pine, norway spruce, silver/downy birch; Tarvainen et al. 2007). Also some sesquiterpenes were detected with the concentrations close to their quantification limits.

Even with relatively high emissions, sesquiterpene concentrations in ambient air remain low due to their high reactivity and very short lifetimes in the atmosphere (Hellén et al. 2018). The main sesquiterpene was β-caryophyllene, which has been detected previously in the emissions of the main tree species in Finland (scots pine, norway spruce, silver/downy birch; Hakola et al. 2001, 2006, 2017; Hellen et al., 2021).

Both mono- and sesquiterpenes had similar diurnal variation with the highest concentrations measured during

early morning hours. In general, the emissions of bVOCs from the vegetation follow the variations of temperature and light being highest in the afternoon (e.g. Hakola et al. 2017; Hellén et al. 2021), however, for these highly reactive compounds with short atmospheric lifetimes, mixing has very strong effect on the local concentration levels. Due to much lower mixing layer with lower dilution during night-time, higher night-time concentrations have been observed for bVOCs (Mogensen et al. 2011; Hellén et al. 2018). In this study, the morning peak of

bVOCs is expected to be a balance between the emissions and mixing. In addition to biogenic emissions, terpenes have some anthropogenic sources. Personal care products and cleaning agents are known to be a source of





especially limonene (Claflin et al. 2021), which was detected also here with the average (± stdev) concentration of 0.054 (± 0.063) μg m⁻³.

### 3.2 Sources of submicron organic aerosol

**3.2.1 Primary OA**

The sources of OA were investigated by the PMF analysis. PMF extracted six different types of OA at the street canyon (Fig. 2) of which three can be considered as POA (HOA-1, HOA-2 and CoOA). HOA-1 had a contribution of 14 % to OA. The mass spectrum of HOA-1 was very similar to that from engine emissions having the largest signal for the hydrocarbon ions $C_4H_9^+$, $C_3H_7^+$, $C_4H_7^+$, $C_3H_5^+$, $C_5H_9^+$ and $C_5H_{11}^+$ at mass-to-charge ratios (m/z's)

57, 43, 55, 41, 69 and 71, respectively (Canagaratna et al., 2004). HOA-1 also displayed a similar diurnal trend with the traffic-related components $BC_{ff}$ and nitrogen oxides (Fig. 3) having the maximum in the morning, however, $BC_{ff}$ increased more sharply in the morning, whereas HOA peaked later at ~9:00. Also, the evening rush hour peak detected for $BC_{ff}$ and $NO_x$ (15:00-17:00), was not apparent for HOA-1. Overall, HOA-1 correlated only moderately with $BC_{ff}$ (Pearson R = 0.52) and $NO_x$ (R = 0.53). Also the total concentration of aVOCs correlated

with HOA-1 (R = 0.67). Both species peaked during morning rush hour and in the evening, but relative to the morning peak, aVOCs had larger concentrations in the evening than HOA-1.

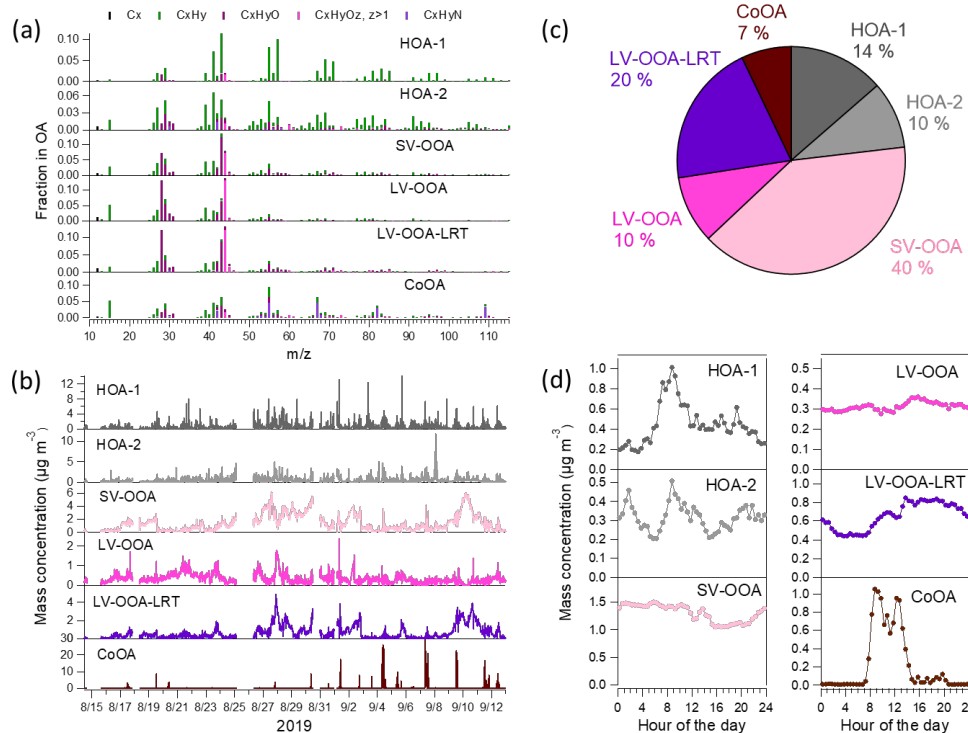

**Figure 2**. Mass spectra (a), time series (b), campaign-average mass fractions (c) and campaign-average diurnal trends (d) for six OA PMF factors at the street canyon.



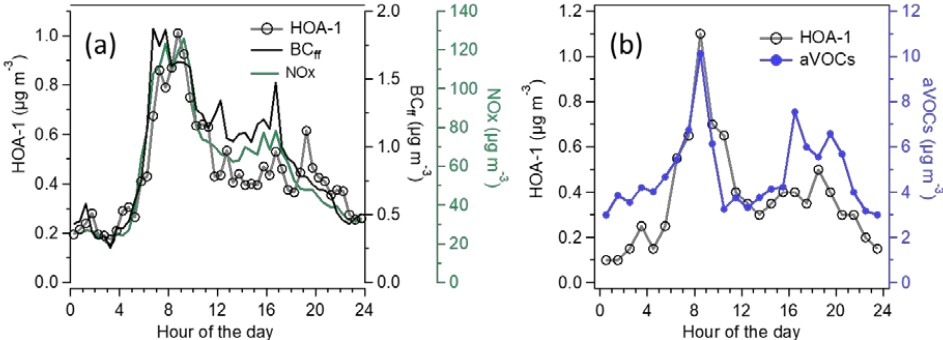

**Figure 3**. Campaign-average diurnal trend of HOA-1, $BC_{ff}$ and $NO_x$ (a), and the diurnal trends of HOA-1 and aVOCs averaged over the VOC measurement periods (b).


HOA-2 had a smaller contribution to OA than HOA-1 (10 %) and it was more oxygenated than HOA-1. Also HOA-2 also had a clear pattern for the hydrocarbon ions, but different from HOA-1, HOA-2 had more signal at lower m/z's, the largest signal being for $C_3H_5^+$, $C_3H_3^+$, $C_4H_7^+$ and $C_3H_7^+$ at m/z's 41, 39, 55 and 43, indicating more double bonds in the hydrocarbon ions and therefore being less saturated. The mass spectrum of HOA-2 also

had a distinct signal for the oxygenated ions $C_2H_4O_2^+$ and $C_3H_5O_2^+$ at m/z's 60 and 73, respectively, that are usually considered typical ions for biomass burning OA (BBOA; Alfarra et al., 2007). In this study, these fragments are unlikely to be related to biomass combustion, as similar to HOA-1, HOA-2 peaked in the morning between 8:00 and 9:00, however, the morning peak was smaller for HOA-2 than for HOA-1. At night-time, the concentrations of HOA-1 and HOA-2 were rather similar. HOA-2 correlated moderately with $BC_{wb}$ (R = 0.41),

but the correlation was stronger for HOA-1 and $BC_{wb}$ (R = 0.59) since also $BC_{wb}$ had a clear morning peak. This is possibly explained by the fact that the aethalometer model cannot resolve $BC_{wb}$ and $BC_{ff}$ completely. Furthermore, traffic emits also carbon which absorbs at near-ultraviolet and lower-visible wavelengths (brown carbon) which can be attributed to $BC_{wb}$ in the aethalometer model, regardless of its original source. The contribution of biomass burning to OA and eBC was likely to be very small since the measurements were carried

out in late summer when ambient temperature was still quite high, and the site was located in the area with apartment buildings with no wood stoves or wood heated saunas.

The relationship of HOA-2 and HOA-1 with traffic species could indicate that the two factors were related to the emissions from different types of vehicles. HOA-2 peaked clearly in the night between Saturday and Sunday (Fig. S5). According to $BC_{ff}$ and $NO_x$ (especially $NO_2$), there were traffic-related emissions during that night, but their

concentrations were not elevated significantly. In the previous studies, it has been shown that for example the exhaust emission of diesel-electric hybrid and ethanol buses equipped with exhaust after-treatment systems can contain $C_2H_4O_2^+$ (m/z 60) and $C_3H_5O_2^+$ (m/z 73) ions in their mass spectra (Saarikoski et al., 2017), however, ethanol buses are practically not in use in the Helsinki area anymore. In terms of particle number, especially the size fraction of 10–25 nm increased particularly during the night between Saturday and Sunday. Järvinen et al.

(2019) have observed a significant fraction of approximately 10 nm sized particles from the retrofitted busses. It can be speculated here that during night-time the vehicle fleet (including e.g. buses and taxis) was newer and equipped with modern exhaust after-treatment systems, causing HOA-2, whereas in daytime, especially in the weekday mornings, there was a large fraction of older heavy duty vehicles producing HOA-1. It should be noted





that SOA produced from the vehicle emissions is also likely to include these oxygenated ions (Timonen et al.,
2017). This is another plausible explanation for the peak of HOA-2 in the night between Saturday and Sunday as
HOA-2 was clearly more oxygenated than HOA-1, however, it has been shown that modern exhaust after-
treatment systems reduce also SOA emissions (Karjalainen et al., 2019). $BC_{wb}$ did not increase markedly during
the night between Saturday and Sunday indicating that the source for HOA-2 was not biomass burning.

Another source for POA at the street canyon was the local coffee roastery. The mass spectrum of CoOA had
pronounced peaks at m/z's 55, 67, 82 and 109 corresponding to the ions of $C_3H_5N^+$, $C_3H_3N_2^+$, $C_4H_6N_2^+$ and
$C_5H_7N_3^+$, respectively, those ions being characteristic for caffeine in the AMS mass spectra (Timonen et al. 2013).
As it can be seen from the time series, CoOA was detected very sporadically, while most of the time its
concentration was near zero. On average, CoOA comprised 7 % of total OA, but during its maximum
concentrations, its contribution to OA was as large as 80 %. Regarding diurnal trends, OA from the coffee roastery
was detected mostly between 7:00 and 14:00, which agreed with the operation hours of the roastery. CoOA did
not correlate with any of the inorganic SP-AMS species. CoOA has been observed earlier in Helsinki at the
SMEAR III station (1 % of OA; Timonen et al. 2013), but compared to SMEAR III, its concentration and
contribution was much larger at the street canyon site because it was much closer to the coffee roastery (street
canyon site is ~600 m north from the roastery vs. SMEAR III is ~1.5 km northeast from the roastery).
The concentrations of CoOA might have been overestimated in this study. Compared to the $PM_1$ mass from the
DMPS, the sum of eBC from the aethalometer and the SP-AMS species (excluding rBC) was clearly larger when
the coffee roastery emissions dominated OA (Fig. S6). This is likely due to the larger relative ionization efficiency
for organics in the coffee roastery emissions since a constant RIE value (default 1.4) was used for organics
regardless of the composition. For LV-OOA, the impact of RIE was opposite to CoOA as $PM_1$ from the SP-AMS
and aethalometer was smaller than that from the DMPS when the LV-OOA fraction had the largest values. For
the other PMF factors, the impact of RIE was less clear. Another reason for higher $PM_1$ from the SP-AMS and
aethalometer could be the enhanced collection efficiency in the SP-AMS. A constant CE of 1 was used in this
study, but the collection efficiency calculated by the Middlebrook et al. (2012) resulted in a CE varying in the
range of 0.45–0.65. Even so, the CE did not seem to explain the difference in $PM_1$ between the DMPS and the
sum of the SP-AMS and aethalometer (Fig. S7).

### 3.2.2 Secondary OA

The largest fraction of OA (78 %) consisted of three types of oxygenated OAs that were likely to be related to
SOA. Two of the OOA factors had very similar mass spectra with the largest signal for the $CO_2$-related ions $CO_2^+$
and $CO^+$ at m/z's 44 and 28, respectively, and the ion $C_2H_3O^+$ at m/z 43. Based on their mass spectra, these OOAs
were classified as LV-OOAs. However, the time series of two LV-OOAs differed clearly as one of the factors had
more stable concentrations throughout the measurement period, whereas the other one increased clearly at the end
of the measurement campaign when the air masses came from Central Europe and Russia (9–11 September 2019;
Salo et al., 2021). Therefore, this LV-OOA is called as LV-OOA-LRT. LV-OOA-LRT had a strong correlation
with inorganic species sulfate (R = 0.87) and ammonium (R = 0.81), while the corresponding correlations with
LV-OOA were less significant (R = 0.35 and R = 0.53, respectively.) LV-OOA had a rather flat diurnal trend,
while LV-OOA-LRT had smaller concentrations from 2:00 to 10:00 than at the other times of the day. The
contributions of LV-OOA and LV-OOA-LRT to OA were 10 and 20 %, respectively.





The third OOA factor was classified as semi-volatile OOA as it had the largest signal for the ion $C_2H_3O^+$ at m/z 43 followed by the $CO_2$-related ions. Of all six PMF factors, SV-OOA had the largest campaign-average

contribution to OA with a fraction of 40 %. The SV-OOA concentration was smaller from 11:00 to midnight than at the other times of the day, similar to nitrate, suggesting its semi-volatile character. Additionally, SV-OOA had a small increase around 14:00 in the afternoon that could be due to the SOA formation in the afternoon. SOA formation will be discussed later in more detail.

Of the six PMF-factors, LV-OOA-LRT was clearly the most oxygenated factor and had the largest oxidation state

(Table 1). That was expected as long-range transported OA has already spent several days in the atmosphere and was exposed to the oxidants before arriving in Helsinki. Also LV-OOA was rather highly oxygenated whereas SV-OOA was much less oxygenated. As anticipated, primary OA sources were the least oxygenated factors. The same pattern was observed when the PMF factors were placed in the O:C, H:C space (Van Krevelen diagram; Fig 4); POA factors were located at smaller O:C and larger H:C values than the OOA factors.


**Table 1**. Elemental ratios, oxidation states and $NO^+$ to $NO_2^+$ ratios for the PMF factors.

| PMF factor | Elemental ratios | | | Oxidation state[a] | $NO^+/NO_2^+$ |
|---|---|---|---|---|---|
| | O:C | H:C | N:C | | |
| HOA-1 | 0.082 | 2.05 | 1.54 e-4 | -1.89 | 1.80 |
| HOA-2 | 0.160 | 1.76 | 0.014 | -1.44 | 1.88 e5 |
| SV-OOA | 0.500 | 1.64 | 4.00 e-3 | -0.640 | 5.85 |
| LV-OOA | 0.710 | 1.60 | 5.20 e-3 | -0.180 | 63.4 |
| LV-OOA-LRT | 0.760 | 1.50 | 1.10 e-2 | 0.020 | 7.92 e-6 |
| CoOA | 0.250 | 1.86 | 9.20 e-2 | -1.36 | 1.88 |

[a]calculated as 2 * O:C – H:C

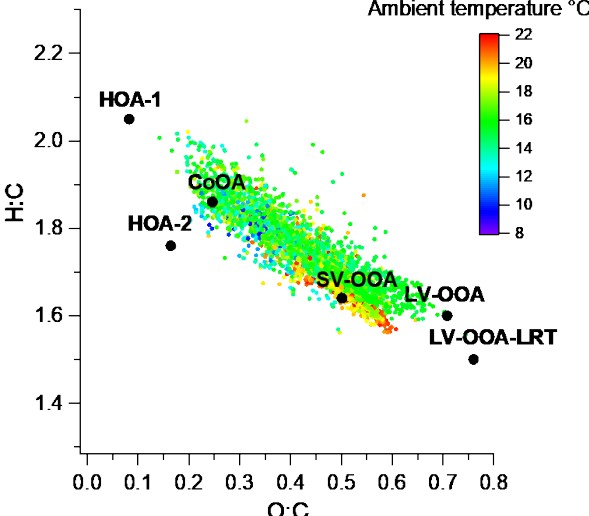

**Figure 4.** Location of OA and PMF factors at O:C and H:C spaces. 1-hour average OA values were colored according to ambient temperature.





### 3.3 Air quality case studies

In order to explain more thoroughly the impact of specific sources on VOCs and particulate species at the street canyon, four periods with different dominating sources were selected for a detailed study. The cases, (1) traffic, (2) coffee roastery, (3) LRT and, (4) biogenic organics, were selected based on the PMF results, VOC and NO/NO$_x$ concentrations, time of the day and ambient temperature. The exact time periods and the criteria for selecting the periods are given in Table S2. Shortly, the traffic period consisted of data from six mornings based on the high NO$_x$ and NO concentrations (NO$_x$ >160 and NO >70 µg m$^{-3}$). The coffee roastery case included a short time period on 7 September (8:10 to 13:40), and the LRT case contained a two-day period from the morning of 9 September to the morning of 11 September. For the biogenic organics, the data measured at ambient temperature > 20 °C was selected, since the bVOC emissions from vegetation are known to be temperature dependent (Tarvainen et al. 2007), but at the same time, the concentration of aromatics needed to be less than 3 µg m$^{-3}$ to exclude the traffic influence.

### 3.3.1 Traffic

As expected, the traffic period was characterized by high concentrations of engine exhaust related pollutants such as NO, NO$_2$, and BC$_{ff}$ (Table S3). Regarding the OA fractions from the PMF analysis, SV-OOA and HOA-1 comprised a majority of OA (50 %). The concentrations of aVOCs, as well as many bVOCs (isoprene, monoterpene and sesquiterpenes), were largest during the traffic period. These peak concentrations in bVOCs are likely related to the low mixing layer height and the resulting low dilution. Notable concentrations of bVOCs explain the large concentration of SV-OOA, which will be discussed in Sect. 3.4.2.

The mass size distribution of OA differed noticeably between the cases (Fig. 5a). For the traffic period, the dominant mode for OA was found at ~300 nm, while a second mode at ~100–150 nm was shown by a shoulder of the larger mode. Mass size distributions were calculated also for the unit mass resolution m/z's 57, 109, 44 and 43 being representative for HOA, CoOA, LV-OOA and SV-OOA, respectively (Fig. 5b). During the traffic period, m/z 57 peaked at the mode at ~100–150 nm, while the mode at larger particle size (at ~300 nm) consisted mainly of oxygenated OA indicated by m/z's 43 and 44 located only at the second mode (not shown). That second mode may not be related to the vehicle emissions, at least not directly, and the more likely source for it was regionally or long-range transported OA during the traffic period.

Regarding particle number (> 10 nm particles), the largest total particle number concentration was measured during the traffic period (~2.3*10$^4$ particles cm$^{-3}$; Table S3). Especially the number of particles at 10–25 nm was large for the traffic period (55 %), and regarding the number size distributions (Fig. S8a), the maximum particle number was observed for the smallest measured DMPS particle size. This is in line with the finding by Rönkkö and Timonen (2019) who have reported that the particles produced by traffic are in the smallest size range from 1–100 nm. The abundance of small particles during the traffic period was also noticed in the total mass size distributions calculated from the DMPS number size distribution (Fig. S8b), but the mass peaked at much larger particle size following a similar trend to the OA size distribution.



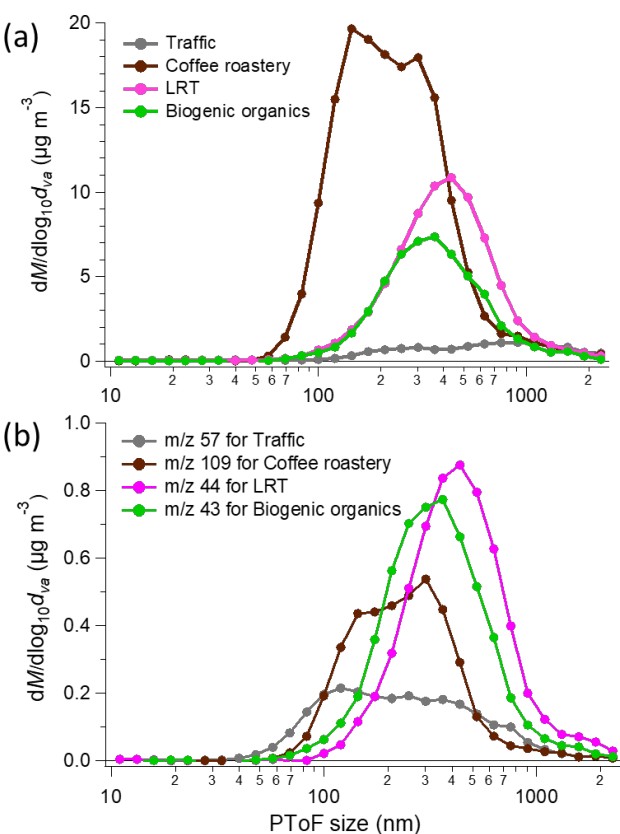

**Figure 5.** Average mass size distributions for OA (a) and selected unit mass resolution m/z's (57, 109, 44, 43) (b)
during four air quality cases (traffic, coffee roastery, LRT, biogenic organics).

### 3.3.2 Coffee roastery

During the coffee roastery period, OA was dominated by CoOA (75 %) and OA was the least oxygenated of all
cases (Table S3). For VOCs, the smallest VOC concentrations were measured during the coffee roastery
emissions. Even though there are gaseous compounds affiliated with coffee roasting, such as sulfur-containing
furfurylthiol (Cerny et al., 2021), they could not be analyzed by the GC-MS method employed in this study. Also,
nitrogen oxide concentrations and $BC_{ff}$ concentrations were small in the coffee roastery period.

The mass size distribution of OA was bimodal peaking at the particle sizes of ~150 and ~300 nm during the coffee
roastery emissions. The dominating mode for coffee roastery OA changed during the 5.5-hour period with the
mode at ~150 nm increasing relative to the mode at ~300 nm when the total concentration of CoOA increased.
M/z 109, a characteristic mass for caffeine, peaked at the second mode (~300 nm) during the coffee roastery
emissions. On the other hand, m/z's 57 and 43 peaked at the first mode (~150 nm) (not shown) suggesting that
the first mode consisted largely of hydrocarbons during the coffee roastery emissions. It is possibly, however, that
the hydrocarbons at ~150 nm are related to the traffic emissions as the coffee roastery emissions were detected



from morning to afternoon when there was also significant amount of traffic. Oxygenated organics (indicated by m/z 44) were split equally between the two modes during the coffee roastery emissions.

The total particle number during the coffee roastery emissions (~2.1*10^4 particles cm^-3) was smaller than that during the traffic period. This difference was especially notable for the small particles, the contribution of 10–25 nm particles to the total particle number being the smallest of all cases (29 %) Accordingly, the coffee roastery
emissions had a maximum at the size of ~55 nm, different from other periods that had the maximum particle number for the smallest measured DMPS particle size. In terms of the mass size distribution, the coffee roastery particles had a bimodal size distribution for the particles < 500 nm, similar to the OA mass size distribution, whereas during the other cases the DMPS mass size distribution was unimodal.

### 3.3.3 LRT

The LRT episode was defined by large concentrations of inorganic species, namely sulfate, nitrate and ammonium. In terms of OA, LV-OOA-LRT made up 31 % of OA, but the contribution of SV-OOA was still larger at 49 %. Out of all cases, OA was most oxygenated during the LRT period. This was expected since the LRT particles have been in the atmosphere for a long time. For the VOCs, the LRT period was not clearly observed due to the oxidation of most VOCs in the atmosphere during the transport, but the ratio of toluene to benzene was the smallest
of all cases. A lower ratio is expected for the long-range transported air masses since toluene lifetime is much lower than the lifetime of benzene in the atmosphere.

The OA mass size distribution mode was around 500 nm, which was the largest mode size of the studied cases. This is an expected result for an LRT period due to the condensation of gaseous species on particles during transport and ageing, increasing their size. Previous studies have also shown that the largest average particle sizes
are observed for atmospherically processed particles that have grown, for instance, during the long-range transport of the airmass (Niemi et al., 2005; Timonen et al., 2008). The mass size distributions of m/z's 44, 43 and 57 were similar in shape to total OA, indicating that both hydrocarbon and oxygenated organics were in the same particles (maximum at ~450 nm). However, for m/z 57 there was a tiny mode also at ~100–200 nm that could be from local traffic. During the LRT period, the particle number was ~ 6–8*10^3 particles cm^-3.

### 3.3.4 Biogenic organics

During the biogenic organics period, the largest fraction of OA (56 %) consisted of SV-OOA followed by LV-OOA-LRT (30 %). Nopinone, which is a tracer for biogenic oxidation, had the largest concentrations during the biogenic organics period, but as already mentioned, most bVOC concentrations were largest during the traffic period due to low mixing. Also ozone concentrations were highest during the biogenic organics period. This could
be speculated to be due to the small NOx, and especially small NO concentrations in the biogenic organics period. Compared to traffic and coffee roastery emissions, the OA mode was at a larger particle size during the biogenic organic emissions (~300–400 nm), but compared to the LRT episode, the mode was at a smaller size. For biogenic organics, m/z's 43 and 44 peaked at a larger size (at ~350 nm) than m/z 57 (at ~300 nm), the mode for m/z 57 broadening to the smaller particle size. In terms of inorganic species, nitrate had very similar mass size distribution
with m/z's 43 and m/z 44, suggesting similar atmospheric processing (local SOA formation), whereas sulfate peaked at much larger particle size (at ~500 nm) indicating its regional or LRT origin during the biogenic organic





period. During the biogenic organics period, the particle number was the smallest of all cases, both in terms of total particle number ($5.8*10^3$ particles cm$^{-3}$) and 10–25 nm particles ($1.8*10^3$ particles cm$^{-3}$).

**3.4 Oxidation of VOCs and SOA formation**

**3.4.1 Oxidation of VOCs**

Oxidation of VOCs under various environmental conditions produces a variety of gas- and particle-phase products that are relevant for atmospheric chemistry and SOA production. To describe this, the production rates of oxidized compounds were calculated from the VOC reactions as described in Sect. 2.3.  The main local sources of oxidation products were $O_3$ oxidation (OxPR$_{O3}$) of bVOCs (66 % of total OxPR) and OH radical oxidation (OxPR$_{OH}$) of

aVOCs and bVOCs (25 % of total OxPR, Fig. 6). OxPR$_{O3}$ stayed relatively constant over the day while OxPR$_{OH}$ was significant only during daytime. This is expected since OH radicals are produced in photochemical reactions only during light hours.  NO$_3$ radical oxidation had only 8 % contribution to total OxPR. In an earlier study at a forest environment in Finland, $O_3$ oxidation was found to be a major oxidation pathway (Hellén et al. 2018), and even with much higher mixing ratios of aVOCs, it was also the case here. Nonetheless, this describes only the

very local situation in the street canyon, and with a bit more regional perspective, the situation may change.

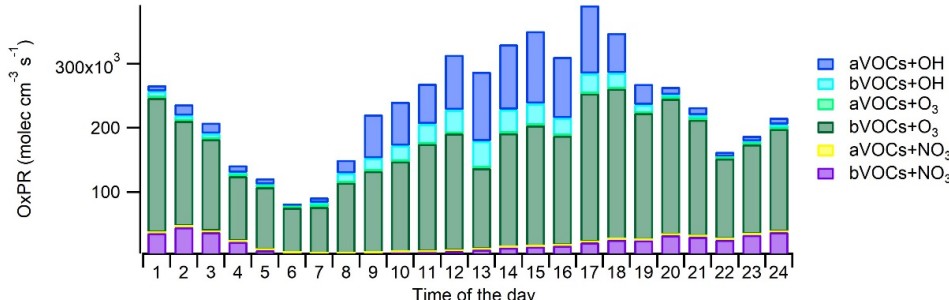

**Figure 6.** Diurnal variation of the production of oxidation products (OxPRs) from the oxidation of anthropogenic VOCs (aVOC) and biogenic VOCs (bVOCs) with hydroxyl radicals (OH), nitrate radicals (NO$_3$) and ozone (O$_3$).


aVOCs produced on average 18 % of the oxidation products at the site and had a major contribution to the OH radical oxidation (72 % of OxPR$_{OH}$). Oxidation of aVOCs with NO$_3$ radicals had a very low contribution (0.3 %) and no contribution to $O_3$ oxidation. The major aVOC for OxPR was p/m-xylene (36 % of aVOC OxPR$_{OH}$) followed by toluene (10 % of the aVOC OxPR$_{OH}$). Even though the contribution of trimethylbenzenes was only

13 % of the aVOC OxPR$_{OH}$, their impact on SOA formation may still be significant due to their higher SOA formation potentials.

bVOCs had 82 % contribution to total OxPR. $O_3$ oxidation was the main oxidation pathway for bVOCs with 82 f contribution. Contributions of OH and NO$_3$ radicals were 9 % and 10 %, respectively. OxPR$_{O3}$ was driven by three monoterpenes, (α-pinene, limonene and terpinolene) and a sesquiterpene (β-caryophyllene) with 20, 13, 33

and 26 % contributions to OxPR$_{O3}$, respectively. For OxPR$_{OH}$, isoprene and α-pinene were the most significant bVOCs. NO$_3$ radical oxidation was driven by four monoterpenes (α-pinene, 3Δ-carene, limonene and terpinolene), which had a 92 % contribution to total OxPR$_{NO3}$. It is clear that bVOCs with lower concentrations and





sesquiterpenes with very low concentrations (~0.004 μg m⁻³) also have a significant effect on local chemistry and due to their high SOA yields (e.g. Lee et al. 2006) possibly also on the SOA formation. During summertime, when

bVOC emissions as well as their ambient air concentrations are higher (Hellén et al. 2012b), their contribution is expected to be even more significant.

**3.4.2 SOA formation**

SOA formation was studied by comparing the diurnal trends of the PMF factors for SOA with the diurnal trends for the modelled OH and $NO_3$ radical concentrations (Fig. 7). LV-OOA had only a small variation throughout the

day, however, the largest concentrations were measured in daytime suggesting that its source is likely to be OH radical reactions with aVOCs. LV-OOA had also a peak during the morning rush hour from 6:30 to 9:00 that was not seen for OH, however, the relative contribution of LV-OOA to OA was smallest in the morning suggesting that the increase was probably due to the low mixing layer height in the morning. Also the advanced exhaust aftertreatment in vehicles can possibly increase the direct emissions of LV-OOA. For instance, the study of Arnold

et al. (2012) indicated elevated exhaust concentrations of organic acids as a consequence of oxidation processes in the exhaust aftertreatment devices. Thus, the distinctive peak during the morning rush hours can be caused also partly by direct emissions of low-volatility organic compounds and their condensation to the particulate phase immediately after the emission. The diurnal trend of SV-OOA differed from that of radical concentrations being smaller in daytime. This indicates that the main factor behind the diurnal trend of SV-OOA was ambient

temperature as SV-OOA is likely to be semi-volatile. LV-OOA-LRT had a slightly larger concentration in daytime than in early morning hours but this was likely to be due to meteorological parameters such as wind direction as LV-OOA-LRT was already highly oxygenated when arrived in Helsinki. On the other hand, the diurnal variation of LV-OOA-LRT had quite a strong correlation with $OxPR_{O3}$ (Fig. S9). Ozone is also strongly related to long-range transport in Finland, but due to the short lifetime of terpenes, this production would be quite local.


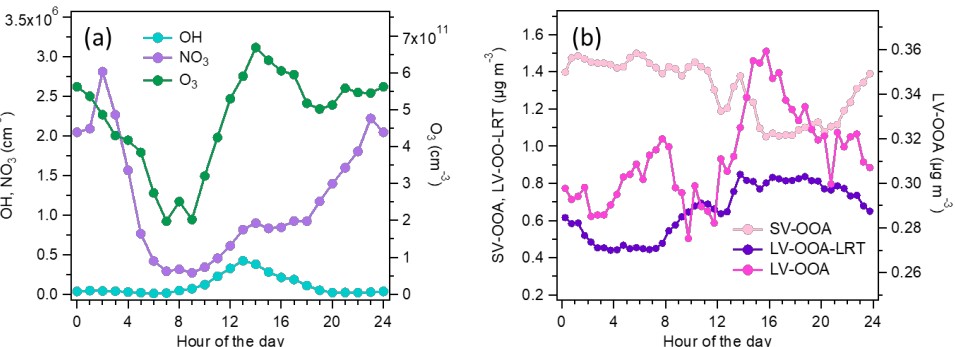

**Figure 7**. Diurnal trends of OH and $NO_3$ and $O_3$ concentrations (a) and the PMF factors related to SOA (SV-OOA, LV-OOA-LRT and LV-OOA) (b).

Additionally, the correlation between VOCs and PMF factors was investigated. Nopinone, an oxidation product of the monoterpene β-pinene, correlated with SV-OOA when the intense long-range transported episode at the end of the measurement period was excluded (R = 0.71). Nopinone is produced in the air through the oxidation of β-pinene by OH (Calvert et al., 2011; Kaminski et al., 2017) and ozone (Grosjean et al., 1993; Hakola et al.,





1994; Winterhalter et al., 2000). The concentration of nopinone is a balance between the production from these
reactions and its own oxidation by OH (Hellén et al. 2018). The main source of β-pinene is expected to be
vegetation, but also some anthropogenic sources are possible (e.g. personal care products or cleaning agents).
The high correlation of the nopinone concentration with the SV-OOA factor supports the fact that SV-OOA is at
least partly related to biogenic emissions. Both nopinone and SV-OOA had maximum concentrations just before
midday (Fig. 8). This can be at least partly explained by the mixing layer height and oxidation rates. During the
night, monoterpenes (and other VOCs) often accumulate in the air due to low mixing layer and lower reaction
sinks (Hellén et al., 2018) as is also seen here by high early morning concentrations. After sunrise, when OH
radical production starts, oxidation rates increase, and more oxidation products (including nopinone) are formed.
The mixing layer height was still relatively low during the morning hours, and nopinone and SV-OOA
concentrations increased. Later during the day when high production of semi-volatile compounds continued,
dilution started to play a role due to the higher mixing layer and the concentrations decreased. This also indicated
the local origin of SV-OOA. Nonetheless, the diurnal variation of SV-OOA and nopinone was relatively small.
This could be explained by the production of semi-volatile compounds from $O_3$ and $NO_3$ reactions of primary
VOCs also during the night. In terms of ambient temperature, OA was located close to the SV-OOA factor in the
Van Krevelen diagram when temperature was high (Fig. 4) which agrees with the larger VOC emissions from
vegetation at higher temperatures.

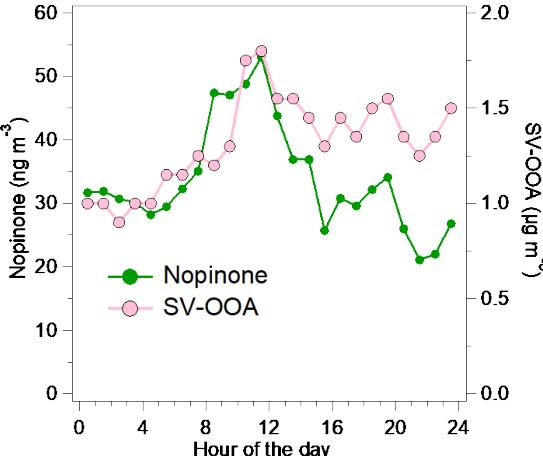

**Figure 8**. Diurnal trends for nopinone and SV-OOA. The intense long-range transported episode at the end of the
measurement period has been excluded from the data.


SOA formation was also studied in terms of organonitrates. In order to investigate secondary organonitrates in the
mass spectra of the PMF factors, $NO^+$ and $NO_2^+$ ions were added to the PMF input data matrix. Both inorganic
and organic nitrate consists dominantly of the ions $NO^+$ and $NO_2^+$, however, the ratio of $NO^+$ and $NO_2^+$ ($NO^+/NO_2^+$)
is different for organonitrates and ammonium nitrate, and therefore allows the determination of organonitrates in
OA (Farmer et al., 2010). Secondary organonitrates are formed mainly during dark aging, via gas-phase $NO_3$
reactions and $RO_2$–NO reactions (Atkinson 2000; Kiendler-Scharr et al., 2016). Fry et al. (2009) have published a
$NO^+/NO_2^+$ value of ~10 for the particles produced in the reaction of a-pinene with nitrate radicals, and Bruns et



al. (2010) have measured a $NO^+/NO_2^+$ values of 10–15 and ~5 for the reactions of nitrate radicals with monoterpenes and isoprene, respectively. In this study, SV-OOA and LV-OOA had larger $NO^+/NO_2^+$ ratios (Table 1) than the PMF factor consisting almost solely of ammonium nitrate fragments in this study (1.2), or pure ammonium nitrate used in the calibration of the SP-AMS (0.8–1.0), suggesting the presence of organonitrates in the SV-OOA and LV-OOA factors. Of the local production of the oxidation products, 8 % was estimated to be from nitrate radical reactions of bVOCs (Fig 6). LV-OOA-LRT had a very low $NO^+/NO_2^+$, which can be explained by the fact that the oxidation products of nitrate radicals can be further oxidized with daytime oxidants (Tiitta et al., 2016). In terms of the POA factors, HOA-2 had a very large $NO^+/NO_2^+$ so it can be speculated that HOA-2 has been oxidized with nitrate radical during the night, however, the contributions of $NO^+$ and $NO_2^+$ were very small in the HOA-2 factor causing a high uncertainty in the ratio. Furthermore, HOA-2 had also the second largest nitrogen to carbon ratio N:C (0.014) after the CoOA factor (Table 1) that included several N-containing ions related to the caffeine (e.g. $C_3H_5N^+$, $C_3H_3N_2^+$, $C_4H_6N_2^+$ and $C_5H_7N_3^+$). As a summary, a schematic diagram depicting the sources and processing of VOCs and OA at the street canyon (Fig S10) is given in supplemental material.

**4 Conclusions**

In this study, state-of-the-art instrumentation was used to measure the concentrations of the main anthropogenic and biogenic VOCs as well as the chemical composition and size distribution of submicron OA at a street canyon to elucidate the sources and features of particulate and gaseous pollutants in urban air. The aim was also to investigate how the VOC data can be used to support the source classification of the AMS data, and the specificity OA source apportionment factors and VOCs to various urban sources was assessed. Furthermore, the production rates of the oxidation products were studied to reveal the main oxidation products and pathways for aVOCs and bVOC in the street canyon environment.

aVOC concentrations were clearly higher than bVOC concentrations at the street canyon. Although the concentrations were lower, the oxidation of bVOCs with ozone was the main local source of oxidation products. bVOCs produced much larger portion of the oxidation products (82 %) than aVOCs (18 %) even though the site was one of the busiest traffic sites in the Helsinki area. Generally bVOC emissions and ambient concentrations are known to be highest during the main growing season in July/early August, but the relatively high ambient temperature during the measurements can at least partly explain the high influence of bVOCs. However, during light hours, OH radical oxidation with aVOCs was also a significant local pathway producing oxidation products. Primary organic aerosol constituted roughly one third of total OA its main sources being traffic and a local coffee roastery. Biomass burning related OA was not observed since ambient temperature was still quite high, and the site was in an area with apartment buildings with no wood stoves or wood heated saunas. On the other hand, secondary organic aerosol, especially from biogenic VOCs as well as from the long-range transport, significantly influenced the OA concentrations. The effect of the main sources was also detected in the mass size distribution of OA as the average size of OA was smaller when POA prevailed compared to the situation when most of OA was made of SOA.

Both VOC and OA data indicated that the dominating sources were traffic and biogenic emissions at the site. The sum of aVOCs correlated well with HOA attributed to the particulate traffic emissions both having a clear maximum during the morning rush hour. The oxidation product of a bVOC, nopinone, correlated with SV-OOA





representing particles originating from biogenic sources. Both nopinone and SV-OOA had maximum concentrations just before midday. The maximum concentration caused by the biogenic sources was observed later than that for traffic related emissions due to different diurnal behaviour of biogenic emissions and local

meteorology. During the night, bVOCs often accumulate in the air due to the low mixing layer height and lower sink reactions as was also seen here by high early morning concentrations. After sunrise, when OH radical production starts, oxidation rates of primary compounds increase, and more oxidation products (including nopinone) are formed. For the biogenic sources, VOCs are important for source classification as the separation and identification of biogenic compounds from the AMS data is challenging due to their extensive fragmentation

and similarity to the other SOA sources.

In contrast, primary OA sources can be separated and identified for the AMS data by using PMF. As shown in this paper, coffee roastery emissions can be explicitly identified from the AMS data due to the unique mass spectrum for caffeine. The gaseous compounds affiliated with coffee roastery activities, for example furfurylthiol, could also be specific for the coffee roastery emissions, however, they could not be extracted with the GC-MS

method selected for this study. Also LRT aerosol was easily separated from OA, the identification supported by inorganic species and air mass trajectories, while its effect was observed less clearly in the VOC measurements due to the oxidation of most VOC in the atmosphere during the transport. Instead, the VOC measurements identified elevated limonene concentrations, associated with cleaning agents and personal care products. This source could not be separated from the OA data, though.

Particle number concentration and size distribution also indicated the main source at some extent. As already shown in many previous studies, traffic related particles were smallest in size, and more than half of the particles were in the size range of 10–25 nm during the traffic emissions. On the other hand, the coffee roastery emissions were noticed to have a maximum in the numbers size distribution at the size of ~55 nm that was different from all the other investigated periods. Particle number concentrations and size distributions are highly relevant for the

health effect studies, especially regarding the ultrafine particles (< 100 nm), however, the VOC and AMS data are much more source-specific than the number concentrations and size distributions. In terms of the chemical composition of the ultrafine particles, this study was lacking a method for the smallest particles as the measurement range of the AMS started at ~60 nm.

Often the identification of VOC and OA sources is challenging in urban areas due to high background

concentrations and complex reactions in the atmosphere. In Helsinki, the pollutant background concentrations are low, and even at the street canyon, the impact of traffic emissions is moderate. This enables detailed identification and characterization of various local sources such as the coffee roastery or limonene associated with human induced emissions. Overall, the results of this study indicate that the use of volatile organic markers complement the sources apportionment of OA. However, proper markers both for gas and particle phases still need to be

identified in order to achieve a comprehensive source analysis for gas and particle phase organics. Another approach could be combining VOC and OA data in the same statistical data analysis method, but the interpretation of the results can be challenging due to e.g. different rates for the atmospheric processes.

*Data availability.* The data shown in the paper is available on request from corresponding author.




*Author contribution.* SaS, JVN, HH, TR and HT designed the experiments and SaS, TT, LMFB and MA performed the measurements. SaS, LP, APP, SiS, PC, RK, LS and AK performed the data analysis. SaS and HH wrote the first version of the manuscript, but all authors participated in the writing process. SaS, TR, HT, HH, APP, SiS, LP and JVN contributed to the acquisition of funding for the study.


*Competing interests.* The authors declare that they have no conflict of interest.

*Acknowledgements.* This work was financed by the European Union's Horizon 2020 Programme Research and Innovation action under grant agreement No 814978 (TUBE) and Academy of Finland projects (Nos 316151, 307797, 323255). This study was also supported by the BC Footprint project (funded by Business Finland, participating companies and municipalities), European Regional Development Fund, Urban innovative actions initiative HOPE (Healthy Outdoor Premises for Everyone, project nro: UIA03-240), MegaSense Growth Engine (Business Finland), Academy of Finland Flagship funding (grant no 337552 and 337551) and COST–COLOSSAL (CA16109). The authors gratefully acknowledge the NOAA Air Resources Laboratory (ARL) for the provision of the HYSPLIT transport and dispersion model and/or READY website (https://www.ready.noaa.gov) used in this publication.

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
