# Peer review of "Characterization of volatile organic compounds and submicron organic aerosol in a traffic environment"

_Atmospheric Chemistry and Physics, 2022_

## Author Comment (AC1)

We thank the Referees for their valuable comments on our manuscript. We made a revision to the manuscript based on the comment of the Referees, and we think that it improved the quality of the manuscript significantly.

Referee comments

**Referee #1**

The authors present measurements at a street canyon in Helsinki, Finland, during late summer. They identify volatile organic compounds (VOCs) in the gas phase that originate from biogenic and anthropogenic sources using online gas chromatography-mass spectrometry (GC-MS). In addition, particle-phase measurements are performed and organic aerosols are further source-apportioned using positive matrix factorization (PMF). The particle size distribution of the different PMF factors is investigated based on the PToF data. Production rates of the oxidized compounds (OxPR) from the VOC reactions are determined using the ARCA box model. Comparison of the PMF factors to the O3, OH, and NO3 concentrations as well as specific VOC oxidation products (e.g. nopinone) highlight the benefit of performing parallel gas and particle phase observations to identify the contribution of gas phase sources to secondary aerosol pollution. This paper fits within the scope of ACP after the following comments are answered.

**Main comments:**

1) Section 3.1: Currently, the paper is missing an overview graph of the location site with info on where the coffee roastery, high traffic, restaurant locations, and other possible sources of emissions are together with the trajectory paths. Furthermore, I believe it would make this section easier to read if the subsections were supported with graphs (even if they are in the supplement).

**Response:** A map showing the location of the measurement site as well as nearby restaurants and the coffee roastery has been added to supplemental material (Fig. S1). Two figures have been added after subsections in 3.1 as suggested by the Referee.

2) Section 3.2: I think we are missing here some back trajectory analysis per factor solution that would further support the origin of the sources. Non-parametric wind regressions and pollution roses would add more confidence to the results for both the primary and secondary OA factors. For example, does the bak trajectory fit the location of the coffee roastery?

**Response:** We don't think that trajectories show best the origin of the OAs as some of the sources are very local but for the LRT episode observed at the end of the campaign (9-11 September 2019), a trajectory plot was added (Fig. S9). However, we added polar plots for the OA factors to present the dependence of OA factor concentrations on wind direction and speed (supplemental material, Fig. S14). According to the polar plots, coffee roastery factor originated clearly from south sector whereas for the other factors, the influence of wind direction and wind speed was less clear. Some discussion on the dependence of OA factors on wind direction have been added.

3) Section 3.4: There are missing VOCs to generate plots like Figures 6 and 7. I understand that there is a limited number of VOCs measured at the canyon but a discussion on the

expected missing compounds would be essential. Did the authors measure carbon monoxide (CO)? If so, emissions could be estimated relative to the emissions of CO based on inventories (see e.g., McDonald et al. 2018 or Coggon et al. 2022). If CO was not measured then benzene could be used as a proxy to determine the emissions from other sources. This could help answer what is the expected influence of many OVOCs that are not measured here and could be a significant fraction of the aVOCs. Is the ARCA box model influenced by the missing VOCs in determining the OH and NO3 concentrations? A sensitivity analysis would be valuable here for the model or further discussion on how these concentrations are determined based on the model.

**Response:** Referee is correct that there are VOCs that were not measured in this study. Therefore, we added a paragraph to Section "Oxidation of VOCs" to discuss on this topic:

"There are also other VOCs e.g. non-methane hydrocarbons and OVOCs including ethanol, acetone, formaldehyde and acetaldehyde etc. in urban air with possibly even higher concentrations (e.g. Hellén et al. 2006) than VOCs measured in this study. Most of them are more volatile and less reactive, and even with high concentrations, their oxidation products are not expected to have significant impacts on local SOA formation. As recent studies on volatile chemical products (McDonald et al., 2018; Coggon et al., 2021; Pennington et al., 2021) show, it is highly probable that there are also other VOCs (e.g. siloxanes and IVOCs), which could contribute to SOA production. They estimate that volatile chemical product (VCP) emissions, which are not traditionally considered as significant VOC source, may be as high as traffic emissions. However, the total OH reactivity measurements in urban ambient air indicate that missing OH reactivity in urban areas has not been this high (Yang et al., 2016). VCP emissions include lots of different compounds, but part of the VCP emissions are aromatics and terpenes, which were also measured in this study and are measured in most total OH reactivity studies. This could also explain why actual missing reactivity in urban air has not been as high as missing VCP emissions."

Also in abstract we defined more clearly VOCs that we analyzed in this study: "The main anthropogenic VOCs (aVOCs, aromatic hydrocarbons) and biogenic VOCs (bVOCs, terpenoids) relevant for secondary organic aerosol formation were analyzed with an online gas chromatograph mass spectrometer"

We also added to conclusions: "Based on the earlier literature (e.g. McDonald et al., 2018; Coggon et al., 2021; Pennington et al., 2021), it is highly probable that there are additional anthropogenic VOCs relevant for SOA formation which were not quantified here, and they may have as high impact on local chemistry as the measured aVOCs. However, even with this much higher aVOC contribution, bVOCs would still be the main source of these oxidation products."

Furthermore, in several places "VOCs" were replaced by "studied VOCs" to indicate that we did not measure all the VOCs found in ambient air.

Regarding CO, even though CO was measured, it is not possible to estimate local concentrations of VOCs based on this due to high variability of lifetimes of VOCs in the air (from minutes to several weeks). Additional problem with benzene is the high background air concentration compared to local concentrations. Of the studied VOCs benzene was the longest living and it complicates its use as a tracer.

In terms of ARCA, we tested the sensitivity of the model by varying VOC concentrations by 20% (uncertainty of our method) and found that [OH] varies by 23% at most and [NO3] by 11% at most (see figure below).

[Figure]

We added to text: "The sensitivity of the model was tested by varying VOC concentrations by 20% (uncertainty of our method) and it was found that [OH] varies by 23% at most and [NO3] by 11% at most."

**Minor comments:**

4)  Line 20: Please define "significantly higher". Isn't this due to the fact that not all anthropogenic VOCs are measured?

**Response:** Even with only aromatic hydrocarbons measured aVOCs were higher than bVOCs. Clarification on the VOC fractions measured was added.

5)  Line 69-71: The authors could add more references here e.g., Coggon et al., 2021, Gkatzelis et al., 2021, and others.

**Response**: Coggon et al., 2021 and Gkatzelis et al., 2021 have been added

6)  Line 83: This could be due to limitations from the AMS detection due to substantial fragmentation that might be important to point out here. Chemical composition could still be different but the AMS would only see fragments and no insights into the functionality of OA.

**Response**: The sentence has been extended to clarify that this observation can be related to the AMS detection: "These results suggest that oxygenated organic aerosol (OOA) components become more chemically similar with photochemical aging regardless of the original source of OOA, however, this finding could be partially caused by the limitations of the AMS detection due to the substantial fragmentation of the chemical species."

In addition, we added a detail (line 82): "the contribution of low-volatility oxygenated organic aerosol (LV-OOA) increased with OA aging when measured with the aerosol mass spectrometer (AMS)" to make it clear that the findings related to OA aging have been obtained from the data measured with the AMS.

7) Line 154: Is a collection efficiency of 1 expected? More discussion and comparison to average values from similar urban studies would be informative here to further validate **this.**

**Response:** A collection efficiency (CE) of 1 was smaller than expected as a CE ~0.7 is typically measured for the SP-AMS. Most likely the larger CE was related to the inaccuracy in the ammonium nitrate calibration. By using the CE=1, the sum of the SP-AMS species (excluding rBC) and aethalometer eBC compared well with $PM_1$ calculated from the DMPS whereas by using CE=0.7 SP-AMS+aethalometer $PM_1$ exceeded $PM_1$ from the DMPS. Typically, CE is calculated according to the equation of Middlebrook et al. (2012) that takes into account acidity and ammonium nitrate fraction of particles as well as humidity. However, for the SP-AMS the CE can be different from the standard AMS due to the additional laser vaporizer. The factors affecting the CE in the SP-AMS have been discussed, e.g., in Onasch et al. (2012), Willis et al. (2014) and Ahern et al. (2016).

A sentence: "The CE of one was larger than that usually calculated for the AMS (Middlebrook et al., 2012) or SP-AMS (Onasch et al., 2012), which could be due to the inaccuracy in the ammonium nitrate calibration" was added.

8) Line 166-169: More discussion on the Aethalometer model would be valuable here or the supplement to avoid readers going to different papers to understand the procedure followed here.

**Response:** More discussion on the aethalometer model as well as the main equations in the model have been added to text.

9) Line 265: I would suggest that inorganic gases are a separate section. Also, adding the standard deviation to the averages would be informative.

**Response:** Results from the inorganic gases have been moved to a separate paragraph. Standard deviations have been added when the concentrations of the species are presented in text.

10) Line 270-271: I would delete this sentence

**Response:** Sentence has been deleted

11) Section 3.1.2: The timeseries of the number size distribution as a 3-D plot for the whole campaign would be informative to have in the supplement when discussing overall trends.

**Response:** Time-series of the 3D dNdlogDp-plot has been added to supplemental material (Fig. S11) as well as diurnal trend for the weekdays.

12) Line 276-277: I would delete this sentence.

**Response:** Based on the other suggestions of the Referee, we assume that the Referee meant the sentence in lines 277-278 "Particle number and mass size distributions will be discussed later in terms of specific sources (Sect. 3.3)." That sentence has been deleted.

13) Figure 1: I would recommend separating this figure into a 4-panel graph. On panel 1 you would have the diurnal variability of aVOCs but adding the standard deviation would provide some more insights into the variability of pollutants. On panel 2 a bar chart of the average concentration of each compound that contributes to the aVOCs from high to low. Panels 3 and 4 would be the same but for biogenic compounds. If there are important correlations to meteorology that explain the bVOC or aVOC trends this could be added to the diurnal profile of panels 1 or 3, respectively.

**Response:** Figure 1 has been modified as suggested by the Referee. Panel (a) has now campaign-average diurnal variation of aVOCs and bVOCs with standard deviations, panel (b) the average diurnal trend of mixing layer height, ambient temperature and solar radiation, and panel (c) and (d) bar charts of the average diurnal variation of specific aVOCs and bVOCs, respectively.

14) Line 326-328: The expected pie chart of compounds that contribute to biogenic emissions and the pie charts of the current observations would be a nice comparison graph. The expected contribution from VCPs as a pie chart is provided by Coggon et al. 2021 and could be compared to the observations too.

**Response:** The contribution of different compounds is now presented in Figure 3. Considering the contribution to biogenic emissions, the contribution of different compounds is highly variable already between different tree species. Therefore, we did not add pie charts on them from earlier studies. It would be a worth of another study to estimate the mean biogenic emissions for this area. In addition, lifetimes of bVOCs are short and vary a lot, and therefore ambient air contributions may be very different to those measured directly from the emission.

15) Line 377-394: Could the HOA factors be influenced by cooking emissions? Are there restaurants or other expected cooking sources nearby the measurement site? Could the cooking OA show similar mass spectra as traffic?

**Response:** We don't believe that HOA was affected by cooking. Cooking factor (COA) has not been found in any of the previous PMF analysis performed for that site (Saarikoski et al., 2021; Chen et al., 2022). It is true that there are restaurants near the measurement station, however, they are all located in the basement of apartment buildings without any outdoor dining areas or barbeque. The exhaust from the restaurants is directed to the roof of the building (mainly 4-5 floor buildings) and therefore it is not expected to affect the measurements at the street level.

However, to investigate this further, Cooking OA has typically substantial signal from oxygenated ions whereas HOA mass spectra have almost exclusively hydrocarbon ions. Cooking factor can be estimated in ambient datasets by investigating the mass-to-charge ratio (m/z) 55 and 57, or ion pairs $C_4H_7^+$ and $C_4H_9^+$ or $C_3H_3O^+$ and $C_3H_5O^+$. In that diagnostic, HOA data aligns to the right arm of the dataset and COA data points aligns to the left arm of the V-shape pattern (Mohr et al., 2012).

We performed COA analysis for our data set and observed that most of the data points are located close to HOA with no clear dependence on the hour of the day. However, we noticed that when coffee roastery OA has a large contribution to OA, data points are located closer to the cooking factor suggesting that coffee roastery OA could be mistakenly interpreted as cooking factor OA if there is no additional data (e.g. wind direction data, reference mass spectra) and prior knowledge of the local sources.

Cooking factor diagnostic plots have been added to supplemental material (Fig. S17) and a paragraph discussing the cooking factor have been added to the end of Section "3.4.1 Primary OA".

Line 404: Are there any back trajectory correlations that support these results?

**Response:** Polar plots calculated for the PMF factors (Fig. S14) show clearly that coffee roastery OA is related only to the south sector where the coffee roastery is located. Additionally, CoOA mass spectra is very unique that resembles closely to that of caffeine (NIST EI spectra, Carbone et al., 2014).

16) Line 432-433: I would delete this sentence.

**Response:** Sentence has been deleted

17) Line 464-465: Does this mean that the traffic emissions are not really only traffic but a combination of traffic and biogenic? It might be worth changing the naming then or making clear that the biogenic emissions only affect the secondary OA production which is still a significant fraction of the mass in this event.

**Response:** "Air quality case studies" section has been deleted from the manuscript as Referee #2 suggested significant changes to the structure of the article.

18) Line 476-477: This is not only the case for traffic but for all factors. Such high particle concentrations at the lower range seem interesting. Could the authors produce a 3-D dNdlogDp timeseries plot that is usually used for "banana" plots? This could inform the readers more regarding the distribution of particles throughout the whole campaign at this urban site. Would such distribution be expected and does it compare well to previous studies?

**Response:** 3D dNdlogDp-plot has been added to supplement (Fig. S11). It is very similar to what has been found earlier at the same site (Barreira et al., 2021). We also added the 3d plot of dNdlogDp in terms of diurnal variation for the weekdays. That figure clearly shows the impact of traffic for the particle size < 50 nm.

We added the sentence: "In general, the number size distribution compared well with the previous studies carried out at the site (e.g. Barreira et al., 2021)."

19) Line 527: Replace "biogenic" with "β-pinene".

**Response:** "biogenic" replaced with "β-pinene"

20) Line 528-531: Is this true also for this period? Are the aromatics high during the event of the biogenic organic? If traffic is high why is NO low? Also, O3 will be high at higher temperatures due to more intense chemistry that could be discussed here.

**Response:** "Air quality case studies" section has been deleted from the manuscript.

21) Line 588: I would recommend avoiding phrases like "quite a strong correlation" with the actual statistical indicators like R and R2

**Response:** "quite a strong" has been removed and a correlation coefficient of R= 0.71 has been added.

22) Figure 7: Is O3 currently constrained in the ARCA box model? If it wasn't would it be captured correctly? Could this be a method to evaluate the model's performance in predicting radical concentrations?

**Response:** Yes, O3 is constrained in the ARCA box model. Without constraining O3 in the model, it is underestimated (see figure below). This could be due to a missing source of O3 during the day (e.g. long-range transport) and the fact that the model does not take into account meteorology. For these reasons, O3 is constrained in the ARCA box model to estimate OH and NO3. We do not think that not constraining O3 in the model is a reliable method to evaluate the model's performance to predict radical concentrations."

[Figure]

23) Figure 8: I would recommend adding b-pinene to the graph since this is the precursor together with OH and O3

**Response:** Due to higher concentrations of β-pinene, we feel that adding β-pinene into Figure 11 (formed Figure 8) would make it unclear for its purpose, which is to show the resembling variation of the oxidation product (nopinone) and SV-OOA. However, β-pinene has the same diurnal variation as the monoterpene sum, which was now added in Figure 3. We added into the text reference to the Figure 3 and clarification that β-pinene follows this variation.

24) Line 629-632: This sentence is hard to read and hard to conclude what the authors want to highlight. Also, please provide numbers in the text.

**Response:** The $NO^+/NO_2^+$ ratio values have been added to text and the sentence has been clarified by separating it into two sentences. Values for the $NO^+/NO_2^+$ ratios have been added.

25) Line 639-641: I would delete this given that the overview graph doesn't say much about what is discussed in this section.

**Response:** This sentence and the related figure in the supplement (Figure S10) has been removed.

26) Line 653-654: Isn't this because the authors are only including a fraction of the anthropogenic VOCs?

**Response:** It is true that we did not measure all anthropogenic VOCs. We clarified now in the text that there could be some contribution from other VOCs. See also response to comment 3.

27) Line 665: Provide R or R2

**Response**: R=0.67 has been added to the correlation of HOA and anthropogenic VOCs. Word "well" has been removed from the sentence: "The sum of aVOCs correlated well with HOA"

**Referee #2**

This study investigated the variation and composition of VOCs and SOA in urban street canyons using various observation instruments. The study provides some helpful information about the source apportionment and aging mechanism of SOA, which can be of considerable interest to readers. This manuscript has the potential to be published in the ACP. However, it has severe structural issues and requires a more sophisticated analysis of the results.

General comments

1) First, the subject of this manuscript is not clear. In Chapter 3.2, the source identification of SOA was investigated, focusing on diurnal variations. In contrast, in Chapter 3.3, the subject suddenly changed to a case study and the focus shifted to the size distribution of aerosols. In Chapter 3.4, the topic changed to the oxidation of VOCs and the generation of SOA. Chapter 3.4.1 analyzed the oxidation contributions of VOCs based on daily variability, while Chapter 3.4.2 focused again on the diurnal variability of SOA oxidation. This flow is discontinuous and disconnected, resulting in a lack of cogency. Therefore, the structure of this manuscript needs to be reorganized extensively, and some content needs to be

omitted. This lack of uniformity also resulted in the abstract and conclusion being extensively long, which needs to be corrected.

**Response:** The structure of the manuscript has been changed comprehensively. Section "3.3. Air quality case studies" has been removed from the manuscript (only few sentences saved and moved elsewhere) to clarify the structure and the subject of the manuscript. Also the order of the other chapters has been changed to make it more fluent. VOC concentrations are now presented after meteorology and inorganic gases, and all results related to particles are presented after that.

Abstract has been modified and shortened slightly. Conclusions has been shortened significantly.

2) Second, most of the interpretation in this manuscript relies on a PMF analysis. PMF has many advantages, but it does not guarantee unique decomposition, and the basis functions can show substantially different patterns depending on the number of basis. In addition, even if the sum of the variabilities of the basis functions explains the data satisfactorily, individual basis functions may not be scientifically significant. Multiple sources may be included in one base function, or conversely, one source may be divided into several basis functions. Therefore, individual base functions must be interpreted cautiously, and the PMF analysis must be validated. The PMF analysis in this study also included certain critical errors. For example, even though the coffee roaster is 600 m away, the CoOA explains 7% of OA on average, which is 30% of the POA caused by traffic and biomass burning (HOA-1, 2). In some periods, CoOA even constitutes 80% of the total OA. This result is not acceptable from a general point of view. Therefore, it is necessary to accurately diagnose the limitations and issues of the current PMF analysis, and these results must be presented in the paper to increase the reliability of the results.

**Response:** We feel that the PMF analysis is important for the subject of the article even though we somewhat agree with the Referee that there can be shortcomings in the method and the interpretation of the results may depend on the PMF user. Therefore, as suggested by the Referee, we added the results from the PMF validation. PMF results were validated by calculating the results by multiple seeds and a bootstrapping analysis was performed. The results from these analyses have been added to supplemental material (Fig. S3 and S4) and short discussion has been added to "Experimental methods" section. Also a residual analysis was performed to assess the quality of the PMF analysis. Figures from the residual analysis are also shown in supplemental material (Fig. S5) and a short discussion on residuals was added to the manuscript. In general, validation tests showed that the 6-factor solution was very stable and similar results were obtained with several parameters. Also the residual analysis showed that there was only a small amount of unexplained mass, and in terms of the mass spectra, the largest relative residuals were noticed for large m/z's that had only a small amount of signal.

Coffee roastery OA has been detected at the measurement site in previous studies (Saarikoski et al., 2021; Chen et al., 2022) as well as at the other sites in the vicinity (Timonen et al., 2013, Carbone et al., 2014, Kuula et al., 2020). We don't think that the contribution of coffee roastery OA is too large (average 7%) as that is close to what have been detected in the previous studies, and also because the coffee roastery is located only 600 meters from the site and the smell of

coffee can be detected at the site often. Compared to the contribution of traffic-related OA, the impact of traffic on OA may have decreased in previous years with the new engine, fuel and aftertreatment technology increasing the contributions of other OA sources.

Specific comments

1. Title: Why is an urban canyon being considered? The manuscript does not present information on the characteristics of the urban canyon.

**Response**: "street canyon" has been replaced by "traffic" in the title to present better the content of the article. Also "street canyon" has been replaced by "traffic environment" throughout the manuscript.

2. Line 213: What is the concentration of OA in the two excluded periods? If PM and OA concentration information for the excluded period is also provided, it will improve readers' understanding.

**Response:** The average OA and $PM_1$ concentrations during period (1) were 65.3 and 67.1 µg m$^{-3}$, and during period (2) 47.5 and 52.6 µg m$^{-3}$, respectively. These values have been added to text.

3. Chapter 3.1.1: A map of the campaign region should be provided to understand the characteristics of the region, the dispersion of nearby sources, and the vortex structure by canyon geometry.

**Response:** A map showing the location of the measurement site as well as nearby restaurants and the coffee roastery has been added to supplemental material (Fig. S1). The spatial variability of air pollutant concentrations at the measurement site has been investigated previously in Järvi et al. (2023) and therefore it was not included in this paper.

We added to Chapter 2.2.3 Aethalometer, DMPS and auxiliary measurements: " In a previous study of Järvi et al. (2023), they have found that the concentration levels at the street canyon are more affected by traffic rates whereas on surrounding areas meteorological conditions dominate pollutant levels."

and

to Chapter 3.1.3 Volatile organic compounds: "In the previous study conducted at the same site in summer (Järvi et al., 2023), they found a very stable atmosphere mostly at night-time indicating limited vertical mixing, whereas in daytime (between 11:00–14:00) very unstable conditions took place indicating well-mixed lower atmosphere"

4. Chapter 3.1.1: Presentation of diurnal variation and time series of CO concentration is helpful because CO is a primary anthropogenic pollutant and follows the characteristics of a passive tracer owing to its long lifetime.

**Response:** The average diurnal trend of CO has been added to manuscript (Fig. 2), and time-series of CO to supplemental material (Fig. S10).

5. Figure 2(b): What is the correlation coefficient between SV-OOA and LV-OOA-LRT? The time series look very similar. If the correlation coefficient is high, why is this so?

**Response:** Even though the time-series look very similar, the correlation coefficient (R) between SV-OOA and LV-OOA-LRT was only 0.656 and there were no particular time-periods with stronger correlation (see figure below).

[Figure]

6. What is the correlation coefficient between $bb_{ff}$ and $bb_{wb}$? HOA-1 shows a stronger correlation with $bb_{wb}$ than $bb_{ff}$, although the changes in HOA-1 show a close pattern with traffic changes. It looks like $bb_{wb}$ does not represent biomass burning emission.

**Response:** The correlation coefficient (R) between $BC_{ff}$ and $BC_{wb}$ was 0.818 showing that there was a quite strong correlation between those components (see figure below). We agree that at our traffic-influenced measurement site $BC_{wb}$ did not represent biomass combustion. It is mentioned in the text that traffic emits also carbon which absorbs at near-ultraviolet and lower-visible wavelengths (brown carbon) which can be attributed to $BC_{wb}$ in the aethalometer model, regardless of its original source.

[Figure]

7. Line 377: The evidence is fragile. There are no types of cars that suddenly operate only on Saturday night (or day).

**Response:** The paragraph speculating HOA-1 and HOA-2 being associated with different types of vehicles has been shortened significantly. All the sentences related to the Saturday-Sunday night have been removed and Figure S5 has been deleted from the supplemental material.

**References**

Ahern, A.T., Subramanian, R., Saliba, G., Lipsky, E.M., Donahue, N.M., Sullivan, R.C.: Effect of secondary organic aerosol coating thickness on the realtime detection and characterization of biomass-burning soot by two particle mass spectrometers. Atmos. Meas. Tech. 9: 6117–6137, 2016.

Barreira, L. M. F., Helin, A., Aurela, M., Teinilä, K., Friman, M., Kangas, L., Niemi, J. V., Portin, H., Kousa, A., Pirjola, L., Rönkkö, T., Saarikoski, S., and Timonen, H.: In-depth characterization of submicron particulate matter inter-annual variations at a street canyon site in Northern Europe, Atmos. Chem. Phys., 21, 6297–6314, https://doi.org/10.5194/acp-21-6297-2021, 2021.

Carbone, S., Aurela, M., Saarnio, K., Saarikoski, S., Frey, A., Timonen, H., Sueper, D., Ulbrich, I., Jimenez, J.-L., Kulmala, M., Worsnop D., Hillamo, R.: Wintertime aerosol chemistry in sub-arctic urban air. Aerosol Sci. Technol. 48, 312–322, 2014.

Chen, G., Canonaco, F., Tobler, A., Aas, W., Alastuey, A., Allan, J., Atabakhsh, S., Aurela, M., Baltensperger, U., Bougiatioti, A.,  De Brito, J. F., Ceburnis, D., Chazeau, B., Chebaicheb, H., Daellenbach, K. R., Ehn, M., El Haddad, I., Eleftheriadis, K., Favez, O., Flentje, H., Font, A., Fossum, K., Freney, E., Gini, M., Green, D. C.,  Heikkinen, L., Herrmann, H., Kalogridis, A.-C., Keernik, H.,

Lhotka, R., Lin, C., Lunder, C., Maasikmets, M., Manousakas, M. I., Marchand, N., Marin, C., Marmureanu, L., Mihalopoulos, N., Močnik, G., Nęcki, J., O'Dowd, C., Ovadnevaite, J., Peter, T., Petit, J.-E., Pikridas, M., Platt, S. M., Pokorná, P., Poulain, L., Priestman, M., Riffault, V., Rinaldi, M., Różański, K., Schwarz, J., Sciare, J., Simon, L., Skiba, A., Slowik, J. G., Sosedova, Y., Stavroulas, I., Styszko, K., Teinemaa, E., Timonen, H., Tremper, A., Vasilescu, J., Via, M., Vodička, P., Wiedensohler, A.,Zografou, O., Minguillón, M. C., Prévôt, A. S. H.: European aerosol phenomenology – 8: Harmonised source apportionment of organic aerosol using 22 Year-long ACSM/AMS datasets, Environment International, 166, https://doi.org/10.1016/j.envint.2022.107325, 2022.

Coggon, M. M., Gkatzelis, G. I., McDonald, B. C., Gilman, J. B., Schwantes, R. H., Abuhassan, N., Aikin, K. C., Arend, M. F., Berkoff, T. A., Brown, S. S., Campos, T. L., Dickerson, R. R., Gronoff, G., Hurley, J. F., Isaacman-VanWertz, G., Koss, A. R., Li, M., McKeen, S. A., Moshary, F., Peischl, J., Pospisilova, V., Ren, X., Wilson, A., Wu, Y., Trainer, M., and Warneke, C.: Volatile chemical product emissions enhance ozone and modulate urban chemistry, Proceedings of the National Academy of Sciences, 118, e2026653118, 10.1073/pnas.2026653118, 2021.

Gkatzelis, G. I., Coggon, M. M., McDonald, B. C., Peischl, J., Aikin, K. C., Gilman, J. B., Trainer, M., and Warneke, C.: Identifying Volatile Chemical Product Tracer Compounds in U.S. Cities, Environmental Science & Technology, 55, 188-199, 10.1021/acs.est.0c05467, 2021.

Järvi, L., Kurppa, M., Kuuluvainen, H., Rönkkö, T., Karttunen, S., Balling, A., Timonen, H., Niemi, J. V., and Pirjola, L.: Determinants of spatial variability of air pollutant concentrations in a street canyon network measured using a mobile laboratory and a drone, Sci. Total Environ, 856, https://doi.org/10.1016/j.scitotenv.2022.158974, 2023.

Kuula, J., Kuuluvainen, H., Niemi, J. V., Saukko, E., Portin, H., Kousa, A., Aurela, M., Rönkkö, T., Timonen, H.: Long-term sensor measurements of lung deposited surface area of particulate matter emitted from local vehicular and residential wood combustion sources, Aerosol Sci. Technol. 54, 190-202, DOI: 10.1080/02786826.2019.1668909, 2020.

McDonald, B. C., de Gouw, J. A., Gilman, J. B., Jathar, S. H.,Akherati, A., Cappa, C. D., Jimenez, J. L., Lee-Taylor, J.,Hayes, P. L., McKeen, S. A., Cui, Y. Y., Kim, S. W., Gen-tner, D. R., Isaacman-VanWertz, G., Goldstein, A. H., Harley,R. A., Frost, G. J., Roberts, J. M., Ryerson, T. B., and Trainer,M.: Volatile chemical products emerging as largest petrochem-ical source of urban organic emissions, Science, 359, 760–764, https://doi.org/10.1126/science.aaq0524, 2018.

Middlebrook, A. M., Bahreini, R., Jimenez, J. L., Canagaratna, M. R.: Evaluation of composition dependent collection efficiencies for the aerodyne aerosol mass spectrometer using field data. Aerosol Sci. Technol. 46: 258–271, 2012.

Mohr, C., DeCarlo, P. F., Heringa, M. F., Chirico, R., Slowik, J. G., Richter, R., Reche, C., Alastuey, A., Querol, X., Seco, R., Peñuelas, J., Jiménez, J. L., Crippa, M., Zimmermann, R., Baltensperger, U., and Prévôt, A. S. H.: Identification and quantification of organic aerosol from cooking and other sources in Barcelona using aerosol mass spectrometer data, Atmos. Chem. Phys., 12, 1649–1665, https://doi.org/10.5194/acp-12-1649-2012, 2012.

Onasch, T.B., Trimborn, A., Fortner, E.C., Jayne, J.T., Kok, G.L., Williams, L.R., Davidovits, P., Worsnop, D.R.: Soot particle aerosol mass spectrometer: Development, validation, and initial application. Aerosol Sci. Technol. 46: 804–817, 2012.

Pennington, E. A., Seltzer, K. M., Murphy, B. N., Qin, M., Seinfeld, J. H., and Pye, H. O. T.: Modeling secondary organic aerosol formation from volatile chemical products, Atmos. Chem. Phys., 21, 18247–18261, https://doi.org/10.5194/acp-21-18247-2021, 2021.

Saarikoski, S., Niemi, J. V., Aurela, M., Pirjola, L., Kousa, A., Rönkkö, T., Timonen: Sources of black carbon at residential and traffic environments obtained by two source apportionment methods, Atmos. Chem. Phys., 21, 14851–14869, doi.org/10.5194/acp-21-14851-2021, 2021.

Timonen, H., Carbone, S., Aurela, M., Saarnio, K., Saarikoski, S., Ng, N. L., Canagaratna, M. R., Kulmala, M., Kerminen, V.-M., Worsnop, D. R., and Hillamo, R.: Characteristics, sources and water-solubility of ambient submicron organic aerosol in springtime in Helsinki, Finland, J. Aerosol Sci., 56, 61-77, https://doi.org/10.1016/j.jaerosci.2012.06.005, 2013.

Willis, M.D., Lee, A.K.Y., Onasch, T.B., Fortner, E.C., Williams, L.R., Lambe, A.T., Worsnop, D.R., Abbatt, J.P.D.: Collection efficiency of the soot-particle aerosol mass spectrometer (SP-AMS) for internally mixed particulate black carbon. Atmos. Meas. Tech. 7, 4507–4516, 2014.

---

## Referee Report (RR1)

I believe all my concerns have been addressed.

---

## Author Response (AR2)

We thank the referee for her/his comments on our manuscript. We have listed referee's comments below and replied to them.

Referee comments

Comments:

1) Line 466-471: This could also be interpreted as cooking emissions captured during these events. CoOA increases during the morning but also at midday. Based on Fig. S1 the trajectory from the measurement location to the Roastery coincides with the location of 3 restaurants. Are there any measurements of aldehydes during the campaign that are known to originate from cooking (e.g., Klein et al., 2019)? These are also reactive compounds that could play a role in the OH and NO3 determination. If these measurements don't exist then I would highlight the need for the measurement of more VOCs in the future and change the naming of the factor to include cooking.

**Reply:** We agree with the referee that the direction of coffee roastery coincides with the location of several restaurants but we disagree that CoOA originates or includes emissions from cooking. Cooking factor has not been found in any of the previous PMF analysis performed for that site (Saarikoski et al., 2021; Chen et al., 2022) whereas coffee roastery emissions have been reported in several studies in the area (Timonen et al., 2013; Carbone et al., 2014; Saarikoski et al., 2021; Kuula et al., 200; Chen et al., 2022). Therefore, we decided not to change the name of the coffee roastery factor.

Unfortunately, we did not have measurement of aldehydes, but as the referee suggested, we included a sentence for the need of more VOC measurements (added to Conclusions):

"This highlights the need for a wider range VOC measurements as also cooking emissions could be identified with the specific VOCs such as unsaturated aldehydes (Klein et al., 2019)."

2) Section 3.5.1 and lines 274-275: I consider that the model sensitivity is not discussed in detail and could play a key role in the "true" OH and NO3 concentrations. The authors currently vary the concentration of VOCs by 20% based on the uncertainty of the method which is not clear how it is defined. They see an equivalent change of 23% for OH and 11% for NO3 which to me shows that the model is sensitive to such changes. I would consider that the missing aVOCs in this study are far more than 20%. Emission inventories of aVOCs could show what the expected fraction of the missing aVOCs is. Currently, there are missing cooking emissions that are highly reactive but also missing (major) less reactive emissions from other sources that could still play a key role in anthropogenic reactivity.

**Reply:** The overall uncertainty of the online TD-GC-MS method is about 18-25% for the measured monoterpenes and sesquiterpenes as reported in the supplementary material of Helin et al. (2020). We decided to use a unified value of 20% uncertainty to test the model sensitivity. As the model is constrained by the in situ measurements and does not rely on emissions, it captures the situation quite well. Highly reactive compounds (e.g. from cooking as the referee mentions, or diterpenes) have not been measured because they have very likely

reacted already before reaching the sampling site and their oxidation products are likely not highly reactive, which is why missing OH reactivity is small in urban environments (see the review by Yang et al., 2016).

The reference of Yang et al. (2016) was missing from the reference list but has now been added there.

3) Line 559-563: Mcdonald et al., 2018 and Coggon et al., 2022, cited by the authors, show that the reactivity from such sources is more than 50% and that even less reactive compounds can play a key role in the observed reactivity. Furthermore, missing reactivity is also dependent on the measurement location (Europe, China, USA, etc.) with multiple studies focusing on this. I consider adding one citation to conclude on the missing reactivity in urban environments to be rather limited.

**Reply:** The one citation that the referee finds limited is a review relying on results from OH reactivity studies. We have made it explicit in the main text that Yang et al. (2016) is a review. It is possible that the missing reactivity can be explained by volatile chemical products (VCPs), however, these were not detected at the site.

4) Line 649-650: I do agree that the measurement location of this study is dominated by biogenic emissions possibly due to the local trees. However, how do the authors conclude on this statement without proper sensitivity analysis? I would consider changing the anthropogenic emissions by a factor of 2 to 10 in the model as a necessary and easy check in order to validate such statements.

**Reply:** As the model is constrained by in situ observations and does not use emission factors, it is not possible to change anthropogenic emissions. Multiplying measured ambient concentrations by a factor 2 or 10 would depict a different situation that the one we were measuring.

Technical comments:

5) Title: There is a typo but even after fixing the typo I consider the title not easy to read. I am also not sure whether an acronym is appropriate for a title.

**Reply:** acronym has been changed to longer name. Also the title has been changed slightly to be: "Characterization of volatile organic compounds and submicron organic aerosol in a traffic environment

6) I would recommend that the authors find all the times they use "very", "rather", etc., and either statistically quantify such statements or delete them.

**Reply:** most of "very" and "rather" words have been deleted or replaced by exact values or more exact words

7) Line 15: What do the authors mean by features? I would delete it.

**Reply: "**features" has been replaced by "characteristics". Features were referring to the important quality or ability of VOCs and OA whereas characteristics refers now to unique qualities that makes them different from others.

8) Line 29: I would change to "The focus of this research was also on the oxidation potential of the measured VOCs and the association…".

**Reply:** Modified as suggested by the referee.

9) Line 37-39: What do the authors mean by "due to specific VOCs attributed to biogenic emissions"? I would delete or rephrase.

**Reply:** Sentence rephrased to be: "Due to the specific VOCs attributed to biogenic emissions, the influence of biogenic emissions was more clearly detected in the VOC concentrations than in OA."

10) Line 419: Delete "also". This sentence is long and hard to follow.

**Reply:** "Also" deleted. The long sentence was also divided into two sentences.

11) Section 3.5.1: Change the title to "Oxidation of measured VOCs".

**Reply:** Title of Chapter 3.5.1. has been changed as suggested.

References

Carbone, S., Aurela, M., Saarnio, K., Saarikoski, S., Frey, A., Timonen, H., Sueper, D., Ulbrich, I., Jimenez, J.-L., Kulmala, M., Worsnop D., Hillamo, R.: Wintertime aerosol chemistry in sub-arctic urban air. Aerosol Sci. Technol. 48, 312–322, 2014.

Helin, A., Hakola, H., and Hellén, H.: Optimisation of a thermal desorption–gas chromatography–mass spectrometry method for the analysis of monoterpenes, sesquiterpenes and diterpenes, Atmos. Meas. Tech., 13, 3543–3560, https://doi.org/10.5194/amt-13-3543-2020, 2020.

Klein, F., Baltensperger, U., Prévôt, A. S. H., El Haddad, I.: Quantification of the impact of cooking processes on indoor concentrations of volatile organic species and primary and secondary organic aerosols. Indoor Air, 29, 926–942, https://doi.org/10.1111/ina.12597, 2019.

Kuula, J., Kuuluvainen, H., Niemi, J. V., Saukko, E., Portin, H., Kousa, A., Aurela, M., Rönkkö, T., Timonen, H.: Long-term sensor measurements of lung deposited surface area of particulate matter

emitted from local vehicular and residential wood combustion sources, Aerosol Sci. Technol. 54, 190-202, DOI: 10.1080/02786826.2019.1668909, 2020.

Saarikoski, S., Niemi, J. V., Aurela, M., Pirjola, L., Kousa, A., Rönkkö, T., Timonen: Sources of black carbon at residential and traffic environments obtained by two source apportionment methods, Atmos. Chem. Phys., 21, 14851–14869, doi.org/10.5194/acp-21-14851-2021, 2021.

Timonen, H., Carbone, S., Aurela, M., Saarnio, K., Saarikoski, S., Ng, N. L., Canagaratna, M. R., Kulmala, M., Kerminen, V.-M., Worsnop, D. R., and Hillamo, R.: Characteristics, sources and water-solubility of ambient submicron organic aerosol in springtime in Helsinki, Finland, J. Aerosol Sci., 56, 61-77, https://doi.org/10.1016/j.jaerosci.2012.06.005, 2013.

Yang, Y, Shao, M., Wang, X., Noelscher, A.C., Kessel, S., Guenther, A., and Williams, J.: Towards a quantitative understanding of total OH reactivity: A review, Atmos. Environ., 143, 147–161, 2016.